# Regional complexity in enteric neuron wiring reflects diversity of motility patterns in the mouse large intestine

Zhiling Li[1], Marlene M Hao[2], Chris Van den Haute[3,4], Veerle Baekelandt[3], Werend Boesmans[1,5,6]*, Pieter Vanden Berghe[1]*

[1]Laboratory for Enteric NeuroScience (LENS), Translational Research Center for Gastrointestinal Disorders (TARGID), University of Leuven, Leuven, Belgium; [2]Department of Anatomy and Neuroscience, University of Melbourne, Melbourne, Australia; [3]Laboratory for Neurobiology and Gene Therapy, Department of Neurosciences, KU Leuven, Leuven, Belgium; [4]Leuven Viral Vector Core, KU Leuven, Leuven, Belgium; [5]Department of Pathology, GROW-School for Oncology and Developmental Biology, Maastricht University Medical Center, Maastricht, The Netherlands; [6]Biomedical Research Institute (BIOMED), Hasselt University, Hasselt, Belgium

**\*For correspondence:**
werend.boesmans@uhasselt.be
(WB);
pieter.vandenberghe@kuleuven.
be (PVB)

**Competing interests:** The authors declare that no competing interests exist.

**Abstract** The enteric nervous system controls a variety of gastrointestinal functions including intestinal motility. The minimal neuronal circuit necessary to direct peristalsis is well-characterized but several intestinal regions display also other motility patterns for which the underlying circuits and connectivity schemes that coordinate the transition between those patterns are poorly understood. We investigated whether in regions with a richer palette of motility patterns, the underlying nerve circuits reflect this complexity. Using $Ca^{2+}$ imaging, we determined the location and response fingerprint of large populations of enteric neurons upon focal network stimulation. Complemented by neuronal tracing and volumetric reconstructions of synaptic contacts, this shows that the multifunctional proximal colon requires specific additional circuit components as compared to the distal colon, where peristalsis is the predominant motility pattern. Our study reveals that motility control is hard-wired in the enteric neural networks and that circuit complexity matches the motor pattern portfolio of specific intestinal regions.
DOI: https://doi.org/10.7554/eLife.42914.001

## Introduction

The gastrointestinal (GI) tract is of key importance in the control of whole body homeostasis. On the one hand, it serves to take up energy and essential nutrients from ingested foods, on the other hand, it has to protect the host from pathogens and dietary antigens, while still maintaining a fine symbiotic balance with the luminal microbiome. In order to do so the GI tract exhibits different motility patterns, which include peristaltic, accommodating, mixing and segmenting activity that varies not only according to the region along the gut but also to the dietary status (*Said, 2012*). The accurate control of GI motility relies on the activity of different types of neurons present in the enteric nervous system (ENS), a ganglionated neural network located in the wall of the gut (*Furness, 2012*). Through largely unresolved circuits, enteric neurons relay information from the gut lumen to motor neurons that steer the action of intestinal smooth muscle cells resulting in coordinated contractions and relaxations of smooth muscle syncytia.

In contrast to the central nervous system where spatial distribution and function of neurons are often linked, the architecture of the ENS is seemingly chaotic. Recently, Lasrado et al. were able to show that functional ENS units in the small intestine are spatially organized in overlapping clonal clusters (*Lasrado et al., 2017*). However, whether the different motility patterns are hard-wired in the ENS and whether these arise from specific or overlapping and possibly multifunctional (*Wood, 2016*) circuit elements remains elusive. The large intestine executes a variety of different motor patterns including segmental activity, tonic inhibition, antiperistaltic and peristaltic waves (*Smith and Koh, 2017*). It has been demonstrated that especially the proximal colon differs from other parts of the large bowel in that it can generate antiperistaltic waves that mix contents to maximally reabsorb water and electrolytes from the lumen, whilst the distal colon is mainly responsible for propelling the fecal pellet along the large intestine via colonic migrating motor complexes (CMMC) (*James, 2011*). Although neural peristalsis has been studied extensively and the underlying mechanisms are largely resolved (*Tonini et al., 1996*; *Costa et al., 2015*; *Hennig et al., 1999*; *Spencer et al., 2001*; *Smith et al., 2014*; *Bornstein, 2009*) little is known about the relationship between peristalsis and other (emptying) motor patterns. Furthermore, the neurogenic control elements for storage, mixing and the transition between the different colonic motor patterns are still far from being understood (*Smith and Koh, 2017*; *Spencer et al., 2016*). Taking advantage of the clearly different motor capabilities of two these regions, we investigated, whether diverse enteric circuits exist that may reflect the neuronal control of these tasks. To do so, we used live $Ca^{2+}$ imaging and focal electrical stimulation to evaluate the connectivity of large numbers of enteric neurons while simultaneously mapping their physical location within the myenteric plexus. We combined this set of experiments with immunofluorescence labeling and viral vector tracing to analyze neuronal identity, morphology, projection orientation and synaptic complexity within the network.

We found that neuronal connectivity is different in two regions of the large intestine. The neuronal wiring in the proximal colon is clearly more complex than in the distal colon, where a larger fraction of neurons completely depends on cholinergic input. The straightforward wiring of the distal colon reflects its limited portfolio of motility patterns and is consistent with that complexity of functional output scales with complexity of the enteric neural network.

## Results

### Motility and underlying neuronal circuitry in the proximal and distal colon

The proximal and distal large intestine have a different function and display a distinct set of motility patterns. For example, while CMMCs are consistently initiated in the proximal part of the colon, propagating CMMCs do not always travel into the distal colon (*Sasselli et al., 2013*; *Barnes et al., 2014*). Video recordings and spatiotemporal map analysis of colonic motility in vitro clearly show that CMMCs start at a regular frequency ($0.38 \pm 0.03$ min$^{-1}$, N = 4 animals) in the proximal colon (*Figure 1A*). In the distal large intestine these peristaltic contractions are observed less frequently ($0.22 \pm 0.07$ min$^{-1}$) and they are dependent on the presence of luminal content, which is supplied by propagating contractions travelling from more proximal regions. To compare the neuronal circuit complexity underlying the differential motor behavior between the proximal and distal large intestine, we used GCaMP3 (Wnt1|GCaMP3) based $Ca^{2+}$ imaging combined with focal electrical stimulation and tested the response signature and location of all neurons that were functionally connected (directly and synaptically) with a specific stimulus stimulation spot. Myenteric ENS preparations from both regions were imaged with a low magnification (5X) lens to maximize the number of ganglia within one field of view, while still being able to resolve individual GCaMP3 expressing neurons (see *Figure 1* and *Figure 1—video 1–4*). With this imaging configuration (*Figure 1B*), we were able to record from a large population of neurons per field of view, containing 25 ± 2 ganglia for the proximal (from eight myenteric plexus preparations, N = 5 animals) and 34 ± 2 ganglia (from seven myenteric plexus preparations, N = 5 animals) for the distal colon.

Electrical stimulation (300 µsec, 20 Hz, 2 s) was delivered with a focal electrode onto an interganglionic fiber tract. We used a volley of 40 pulses, considered to be a maximal stimulus (*Fung et al., 2017*), to assure that all neurons functionally (both synaptically or directly) connected to the stimulation site were activated. This induced a sharp increase in $[Ca^{2+}]_i$ in myenteric neurons scattered

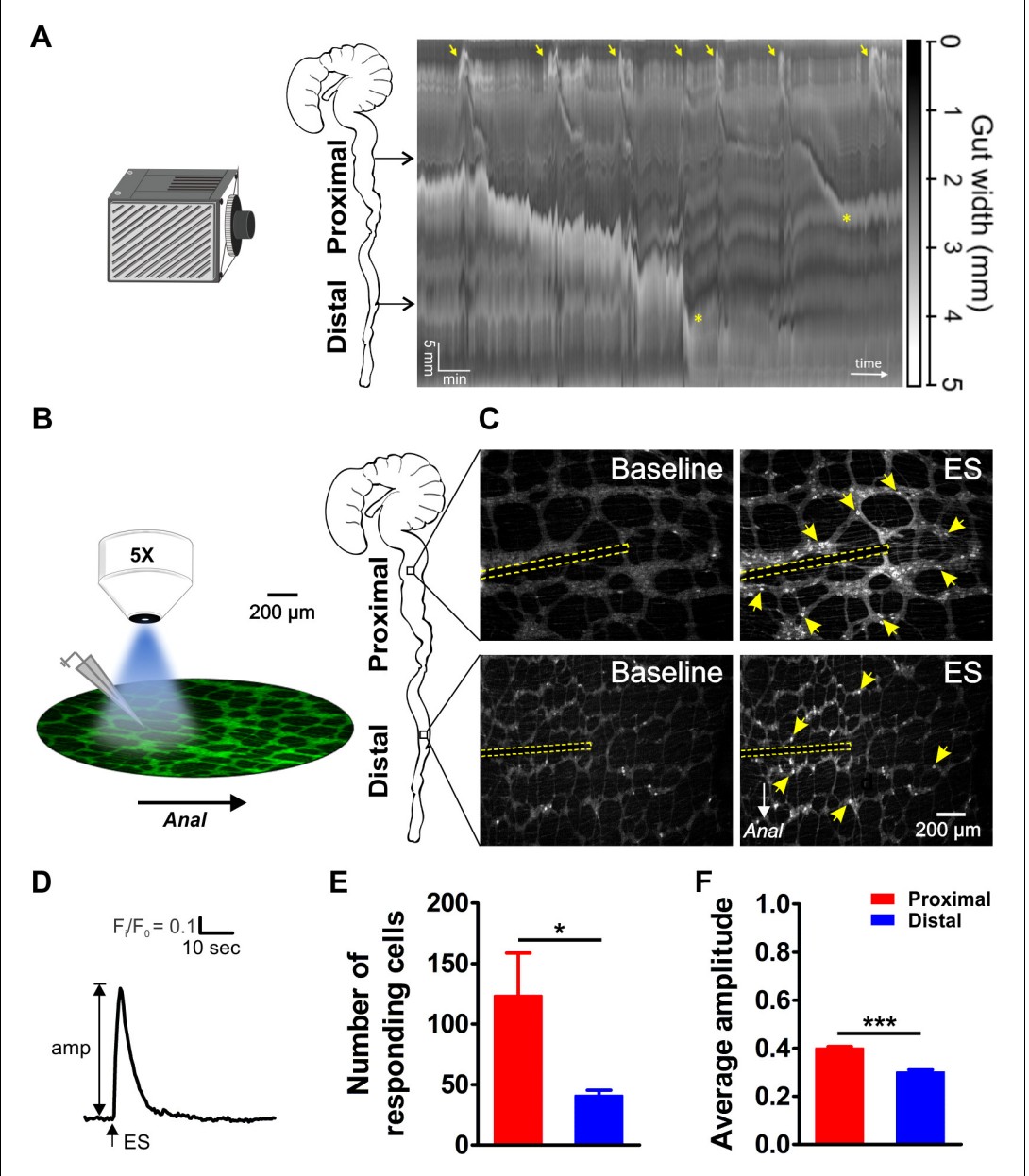

**Figure 1.** Video imaging of colonic motility in vitro and in situ calcium imaging of myenteric neuron activity in the proximal and distal colon. (**A**) Video recordings of the isolated mouse colon (with caecum attached) were analyzed using spatiotemporal mapping of the intestinal diameter (representative example of 4 experiments). Maximum constriction (black), maximum dilation (white), and intermediate levels of constriction (grayscale) of the whole colon (vertical axis: 6 cm in total) are represented over time (horizontal axis, 15 min total duration). Colonic migrating motor complexes (CMMCs) are initiated in the proximal colon (yellow arrows). Propagating CMMCs proceed into the distal large intestine when associated with luminal content supplied by more proximal regions (yellow asterisks). (**B**) Schematic overview of the experimental setup (left) and regions of the mouse large intestine that were compared (right). Colonic myenteric plexus preparations obtained from *Wnt1-Cre;R26R-GCaMP3* (Wnt1|GCaMP3) mice were visualized under an upright fluorescence microscope using a 5X objective lens. Neuronal $Ca^{2+}$ transients were elicited by trains of electrical pulses (300 μsec, 20 Hz, 2 s) transmitted via a focal electrode positioned on interganglionic fiber tracts in the center of the field of view. (**C**) Representative single frames taken from GCaMP3 fluorescence recordings of neurons within myenteric ganglia of proximal (top row) and distal (bottom row) colon before (baseline) and during electrical stimulation (ES, the position of the focal electrode is depicted by the yellow dotted line) (see corresponding suppl. movies). A random subset of responsive neurons is marked with yellow arrows. (**D**) Representative trace of an ES-evoked $Ca^{2+}$ transient of an individual myenteric

*Figure 1 continued on next page*

*Figure 1 continued*

neuron stimulated in the control situation. The amplitude of each $Ca^{2+}$ transient was calculated as the difference between baseline ($F/F_0$) and maximal $F_i/F_0$ GCaMP3 fluorescence. (E) Comparison of the average number of neurons responding per field of view (2.2 mm$^2$) (123.5 ± 35.3 vs 41.0 ± 4.4, *p=0.049). (F) Comparison of the average $Ca^{2+}$ transient amplitude (0.40 ± 0.01 vs 0.30 ± 0.01, ***p<0.001) elicited by fiber stimulation in control. Eight myenteric plexus preparations (N = 5 animals) in the proximal and seven myenteric plexus preparations (N = 5 animals) in the distal colon were used for calculating the data in E and F.

DOI: https://doi.org/10.7554/eLife.42914.002

The following video and source data are available for figure 1:

**Source data 1.** NO. of mice and responding cells.
DOI: https://doi.org/10.7554/eLife.42914.007
**Source data 2.** Responding cells and Ca imaging amplitude.
DOI: https://doi.org/10.7554/eLife.42914.008
**Figure 1—video 1** Stimulaiton one in proximal colon.
DOI: https://doi.org/10.7554/eLife.42914.003
**Figure 1—video 2.** Stimulation two in proximal colon.
DOI: https://doi.org/10.7554/eLife.42914.004
**Figure 1—video 3.** Stimulation one in distal colon.
DOI: https://doi.org/10.7554/eLife.42914.005
**Figure 1—video 4.** Stimulation two in distal colon.
DOI: https://doi.org/10.7554/eLife.42914.006

around the electrode in both the proximal and distal large intestine (*Figure 1C and D*). Both the number (#) of responding (R) neurons (#R) (*Figure 1E*) and the maximal $Ca^{2+}$ transient amplitude ($\Delta F_i/F_0$, *Figure 1F*) were significantly higher in the <u>prox</u>imal compared to the <u>dist</u>al colon (#$R_{Dis}$/#$R_{Prox}$ = 33%).

## Regional differences in myenteric plexus morphology

To investigate why more myenteric neurons in the proximal colon responded to the stimulus, we first assessed whether this difference could simply be explained by differences in the density of neurons. Using immunohistochemistry for the pan-neuronal marker Hu, we found a higher density of neurons (number of neurons: #N) in the proximal colon compared to the distal (*Figure 2A–B*), resulting in a ratio of #$N_{Dis}$/#$N_{Prox}$ = 0.85 (85%). Next, we also quantified the number of neuronal fibers present in one interganglionic fiber tract by staining for neuronal class III β-tubulin (Tuj1) and found that interganglionic fiber tracts in the proximal colon contained more neuronal processes than those in the distal colon (*Figure 2C–D*). Therefore, given that the focal electrode covers the entire width of the interganglionic fiber tract in both regions, a greater number of neuronal fibers is activated in the proximal colon with each electrical stimulus. However, even taken together, the higher density of myenteric neurons and the difference in activated fiber (F) number (#$F_{Dis}$/#$F_{Prox}$ = 0.50 (50%)) cannot fully explain the higher number of responding neurons in the proximal colon (#$N_{Dis}$/#$N_{Prox}$ (85%) * #$F_{Dis}$/#$F_{Prox}$ (50%)=42.5%, which is greater than the observed responder (#R) ratio #$R_{Dis}$/#$R_{Prox}$ = 33%). In a simplistic model where all targeted processes belong to monoaxonal neurons that connect with only one postsynaptic neuron, interganglionic fiber tract stimulation would activate two neurons per fiber: one synaptically and one antidromically. In the distal colon, this oversimplified assumption does not deviate too much from the numbers of responding neurons observed (18 fibers stimulated * 2 = 36 responding neurons), while in the proximal colon this assumption definitely does not appear to hold true (36 fibers stimulated * 2 $\neq$ 123 responding neurons) and suggests that there is far more complex wiring, including an increased number of synaptic targets for each neuron.

## $Ca^{2+}$ response signatures of individual myenteric neurons

To assess the role of individual neurons and their synaptic inputs in the enteric circuitry, we constructed 'activity over time' (AoT) (*Boesmans et al., 2013*) images in which responding cells, color coded by amplitude, can be identified (*Figure 3A*). For each neuron we determined a response signature, which we defined as the ratio of their $Ca^{2+}$ response amplitude in two consecutive rounds of electrical stimulation. As seen in the AoT images, neurons display a variety of response signatures,

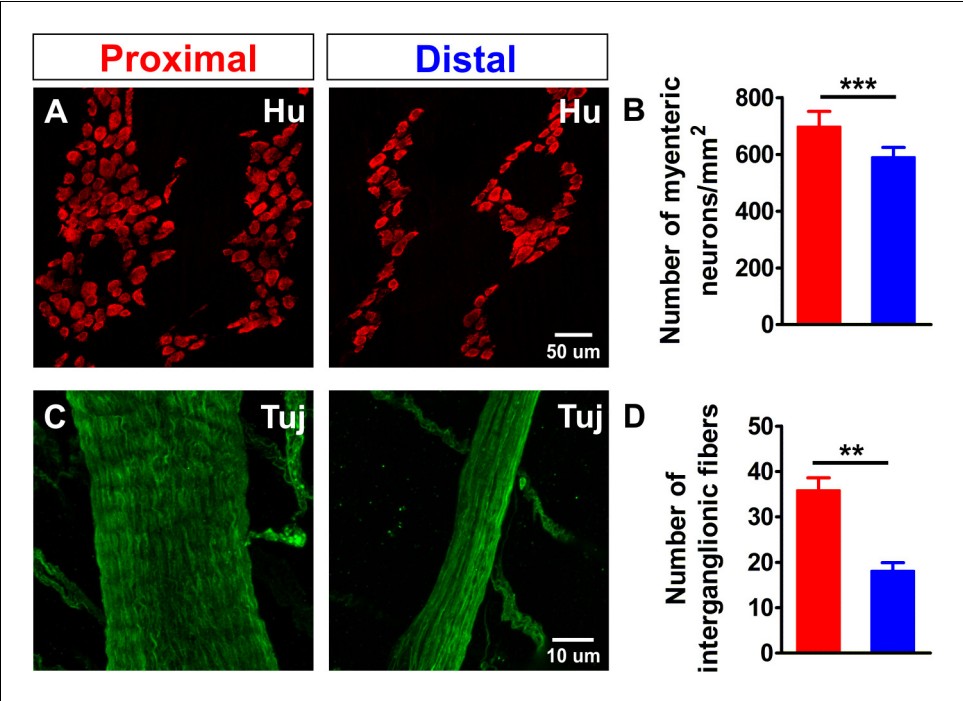

**Figure 2.** Density comparison of myenteric neurons and interganglionic processes in the proximal and distal colon. (**A**) Confocal maximum projections of whole-mount preparations of the myenteric plexus from the proximal and distal colon immunostained for the pan-neuronal marker Hu (red). (**B**) Quantification of the number of myenteric neurons per square millimeter ($698.3 \pm 52.9$ vs $591.3 \pm 33.2$ per $mm^2$, \*\*\*p<0.001; N = 3) in the proximal (red) and distal (blue) colon. The ratio between the number of neurons (#N) in the distal and proximal colon ($\#N_{Dis}/\#N_{Prox} = 0.85$) is 85%. (**C**) Confocal maximum projections of whole-mount preparations of the myenteric plexus from the proximal and distal colon immunostained for neuronal class III β-tubulin (Tuj1, green). (**D**) Quantification of the number of processes per interganglionic fiber bundle ($35.9 \pm 2.7$ vs $18.2 \pm 1.8$, \*\*p=0.005; N = 3), the ratio between the number of fibers in the distal vs. proximal ($\#F_{Dis}/\#F_{Prox} = 0.50$) is 50%.
DOI: https://doi.org/10.7554/eLife.42914.009

The following source data is available for figure 2:

**Source data 1.** NO. of mice.
DOI: https://doi.org/10.7554/eLife.42914.010
**Source data 2.** NO. of myenteric neurons.
DOI: https://doi.org/10.7554/eLife.42914.011
**Source data 3.** NO. of interganglionic fibers.
DOI: https://doi.org/10.7554/eLife.42914.012

with some having increased while others show decreased amplitudes during the second stimulus (over 95% of the neurons responded twice). Based on their response signature, we classified the neurons into one of five types (*Figure 3B*). Type I (black) 'blocked' neurons only responded to the first stimulus; type II (blue), 'reduced' neurons, in which the $Ca^{2+}$ response to the second stimulus was reduced compared to the first stimulus, type III (green), 'unchanged', where both stimuli elicited a similar $Ca^{2+}$ transient; type IV (red), neurons with an 'increased' response; and type V (purple), 'new' cells that did not respond to the initial stimulus but appeared during the second electrical stimulation (*Figure 3B*). The frequency histograms (*Figure 3C*), averaged over different stimulation pairs and preparations, illustrate the consistency of $Ca^{2+}$ responses over consecutive experiments as over 85% of the neurons displayed equal $Ca^{2+}$ transients or were only slightly (>0.8) reduced or increased (<1.2). This consistency is also visible in *Figure 1—Video 1–4*, which show an example of the response to the first and the second stimulation in the proximal and distal colon respectively.

We used this classification scheme to investigate the contribution of cholinergic synaptic activation in the myenteric circuitry of the large intestine. Cholinergic transmission, involving the activation of nicotinic receptors (nAChRs) is established early on in development (*Foong et al., 2015*) and

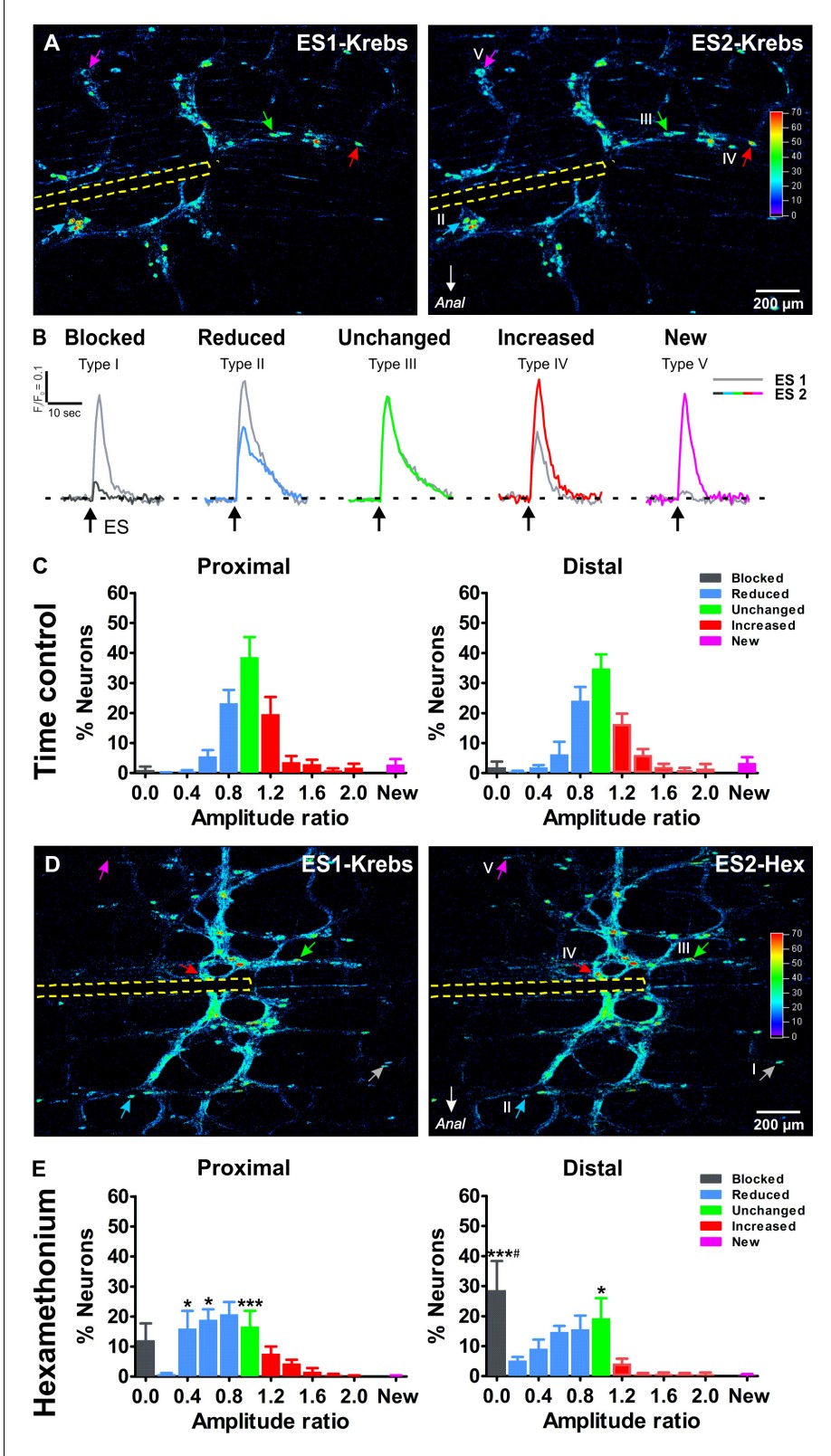

**Figure 3.** Ca²⁺response signatures of enteric neurons during two consecutive rounds of electrical stimulation in control conditions and in Hexamethonium. (**A**) Activity over Time (AoT) images in which the Ca²⁺ transient amplitude for active cells only is color-coded (absolute values in arbitrary units, see color scale). Left and right respectively show an example of the responses (arrows point at individual examples) to a first electrical stimulation (ES1) and a second consecutive electrical stimulation (ES2) in control conditions (Krebs). The location of the focal electrode is indicated by

*Figure 3 continued on next page*

*Figure 3 continued*

the yellow dashed line. Colored-coded arrows mark responder subtypes as explained in B. (B) The amplitude ($\Delta F/F_0$) of the second (color trace) response was compared to the first (gray trace) and expressed as a ratio ($\Delta F/F_0)_{ES2}$ / ($\Delta F/F_0)_{ES1}$ for each individual neuron. Based on this response signature, responsive neurons were classified into five different classes: blocked (Type I), reduced (Type II), unchanged (Type III), increased (Type IV) and new (Type V) cells (see color-coded arrows in A, (A')). Note that in this field of view no Type I neuron was present, as these are very rare in control (saline) conditions. (C) Histograms showing the percentage of neurons (mean ±SEM) belonging to the different (color coded) classes as found in the myenteric plexus of the proximal (left) and distal (right) colon. Results are expressed as the amplitude ratio binned by 0.2. Note that the distributions approximate a standard normal distribution both in the proximal and distal colon do not differ substantially between the proximal and distal colon, indicating a robust response behavior. Data were obtained from Ctrl-Ctrl stimulation pairs in six myenteric plexus preparations (N = 4 animals) in the proximal and six myenteric plexus preparations (N = 4 animals) in the distal colon. (D) Activity over Time (AoT) images in which the maximal $Ca^{2+}$ amplitude (color-coded) of responsive neurons is shown during a first electrical stimulation (ES1) in control Krebs (left) and a second consecutive stimulation (ES2) in the presence of hexamethonium (Hex, 200 µM, right). The location of the focal electrode is indicated by a yellow dashed line. Neurons belonging to each of the five different types of responders classes (blocked (Type I), reduced (Type II), unchanged (Type III), increased (Type IV) and new (Type V)) are indicated by color-coded arrows. (E), Histograms showing the percentage of neurons (mean ±SEM) belonging to the different (color coded) classes in the presence of Hex as found in the myenteric plexus of the proximal (left) and distal (right) colon. These frequency histograms show a shift to the left (more neurons in the blue and black bars) as compared to the control situation. About ~10% of proximal neurons and ~30% of distal neurons were completely blocked by Hex (Prox: 12.2% vs 1.1%, p>0.05; Dis: 28.7% vs 1.9%, ***p<0.001, two-way ANOVA with Bonferroni *post hoc* test). Comparing the Hex effect between both regions, it is clear that the proportion of blocked neurons in the distal is significantly higher than in the proximal colon (Dis: 28.7 ± 9.7% vs Prox: 12.2 ± 5.6%, #p<0.05, two-way ANOVA with Bonferroni *post hoc* test). Data were obtained from Ctrl-Hex stimulation pairs in six myenteric plexus preparations (N = 4 animals) in the proximal and seven myenteric plexus preparations (N = 5 animals) in the distal colon. The * symbols denote the comparison between control and Hex, while # reflects the comparison distal vs proximal.

DOI: https://doi.org/10.7554/eLife.42914.013

The following source data is available for figure 3:

**Source data 1.** NO. of mice.
DOI: https://doi.org/10.7554/eLife.42914.014
**Source data 2.** Ca imaging amplitude ratio of proximal colon.
DOI: https://doi.org/10.7554/eLife.42914.015
**Source data 3.** Ca imaging amplitude ratio of distal colon.
DOI: https://doi.org/10.7554/eLife.42914.016

remains a crucial component of excitatory synaptic transmission in the ENS throughout life (*Galligan and North, 2004*; *Gwynne and Bornstein, 2007*). We used hexamethonium, an effective blocker of the nicotinic acetylcholine receptor, to inhibit cholinergic neurotransmission and refine the wiring identity of individual neurons. As in our control experiments, each preparation was stimulated twice, once in control Krebs and a second time after 10 min incubation in hexamethonium (200 µM; *Figure 3D*). Logically, in the presence of hexamethonium, the frequency histograms shifted to the left (*Figure 3E*) as more neurons were present in the blue bins both in the proximal (56.6% vs 29.6% in control) and distal (44.8% vs 32.8% in control) colon. In line with the reduction in amplitude, also the proportion of blocked neurons (black bin) was higher in the hexamethonium versus the control condition. This proportion of Type I 'blocked' neurons was significantly higher in the distal as compared to the proximal colon (*Figure 3E*), indicating that in the distal colon a larger fraction of neurons completely depends on cholinergic input from the stimulation site. Even though most fast excitatory neurotransmission is blocked by hexamethonium, a considerable number of neurons remain responsive and some even displayed enhanced responses to interganglionic fiber tract stimulation (*Figure 3D*).

## The importance of cholinergic input scales with distance from the stimulation electrode

As many of the blocked neurons were located aborally to the electrode, we hypothesized that the subtle circuitry differences observed in the vicinity of the stimulated interganglionic fiber tract might be more explicit when monitoring the neurons even more distal to the stimulation site. We therefore extended the experiments by imaging one field of view below the original field (see schematic *Figure 4A*).

Indeed, for both the proximal and distal colon, the effect of hexamethonium on inhibition of neural activity was more explicit further away from the stimulation site (*Figure 4B–C*). The average $Ca^{2+}$ transient amplitude of the neurons in this field of view was lower than for neurons closer to the

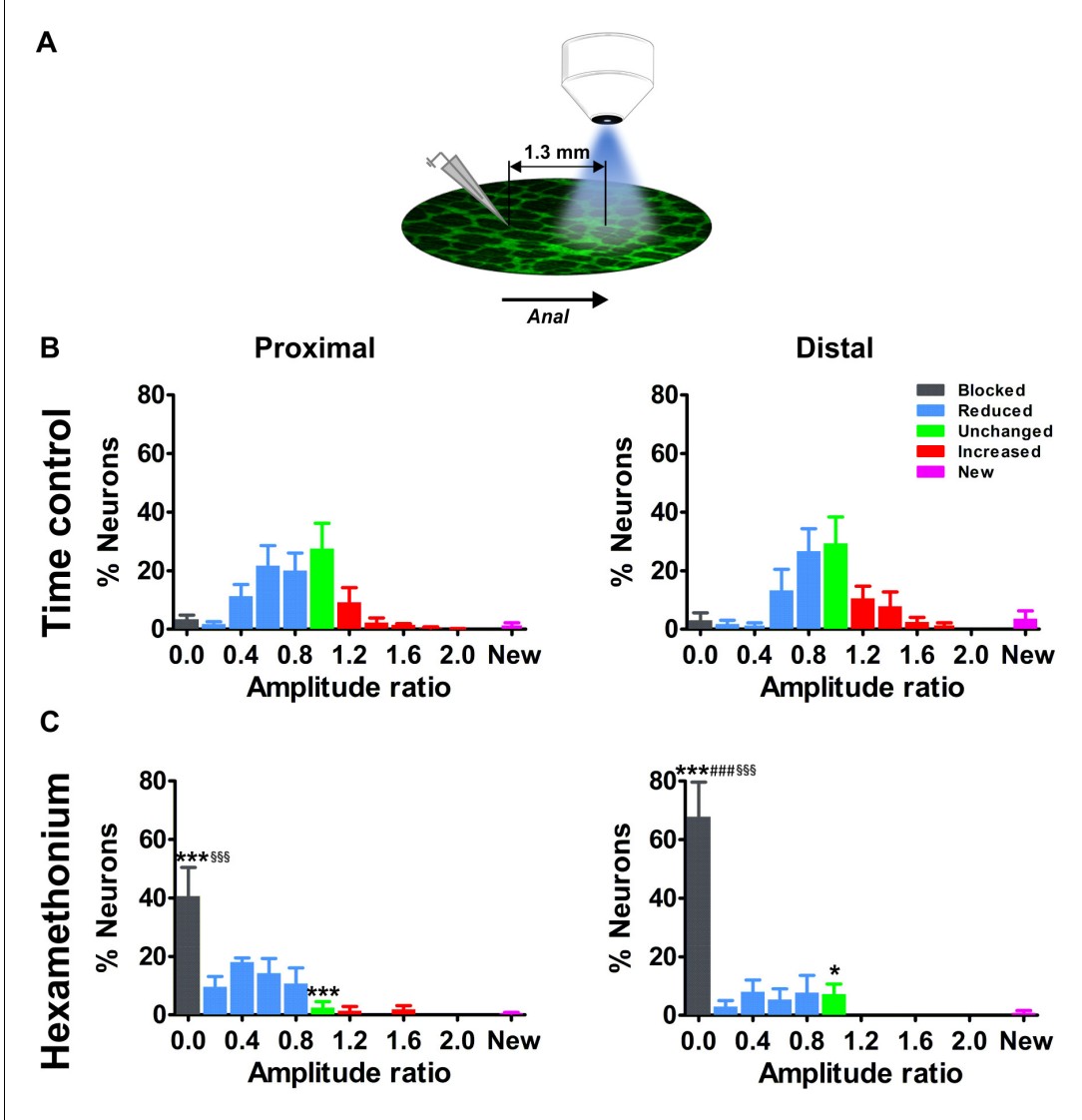

**Figure 4.** Effects of blocking cholinergic nicotinic neurotransmission on electrically evoked $Ca^{2+}$ transients of myenteric neurons distal to the stimulation site. (**A**) Schematic representation of the stimulation site and imaging field. For the current set of experiments, the electrode was placed one field of view (=1.3 mm) orally to the field of view. (**B–C**) Histograms showing the percentage of neurons (mean ±SEM) belonging to the different (color coded) classes as found in the myenteric plexus of the proximal (left) and distal (right) colon at a distance away from the stimulation electrode. (**B**) shows the control condition (two stimuli in control Krebs) and (**C**) the situation when the second stimulus was given in the presence of hexamethonium (Hex, 200 μM). Note that the histograms in (**B**) are still normally distributed but with a larger spread compared to the neurons closer to the electrode (*Figure 3*). Data were obtained from Ctrl-Ctrl stimulation pairs in five myenteric plexus preparations (N = 3 animals) in the proximal and seven myenteric plexus preparations (N = 3 animals) in the distal colon. The frequency histograms in (**C**) show a robust shift to the left (more neurons in the blue and black bars) as compared to the control situation. About ~40% of proximal neurons and ~70% of distal neurons were completely blocked by Hex (Prox: 40.7% vs 3.5%, ***p<0.001; Dis: 67.8% vs 3.1%, ***p<0.001, two-way ANOVA with Bonferroni *post hoc* test). Comparing the Hex effect between both regions, many more neurons were completely blocked in the distal compared with the proximal colon (Dis: 67.8 ± 11.7% vs Prox: 40.7 ± 9.7%, ###p<0.001, two-way ANOVA with Bonferroni *post hoc* test). In addition, comparing the Hex effect between both fields of view, the proportion of blocked neurons was significantly higher in the field further away than close to the electrode (see in *Figure 3E*) (Prox: 40.7 ± 9.7% vs 12.2 ± 5.6%, §§§p<0.001; Dis: 67.8 ± 11.7% vs 28.7 ± 9.7%, §§§p<0.001, two-way ANOVA with Bonferroni *post hoc* test). Data were obtained from Ctrl-Hex stimulation pairs in four myenteric plexus preparations (N = 2 animals) in the proximal and four myenteric plexus preparations (N = 3 animals) in the distal colon. The * symbols denote the comparison between control and Hex, the # symbols reflect the comparison between distal and proximal and the § symbols the comparison between the fields of view (close and further away from the electrode).

DOI: https://doi.org/10.7554/eLife.42914.017

The following source data is available for figure 4:

**Source data 1.** NO. of mice.

*Figure 4 continued on next page*

*Figure 4 continued*

DOI: https://doi.org/10.7554/eLife.42914.018
**Source data 2.** Ca imaging amplitude ratio of proximal colon.
DOI: https://doi.org/10.7554/eLife.42914.019
**Source data 3.** Ca imaging amplitude ratio of distal colon.
DOI: https://doi.org/10.7554/eLife.42914.020

electrode (data not shown). The proportion of fully blocked neurons (type I) was significantly higher compared to the original field of view (*Figure 3E and 4C*). In addition, the fraction of blocked neurons in the distant field (>650 µm) was significantly larger in the distal colon compared to the proximal colon (*Figure 4C*), which indicates that, in the distal large intestine, more neurons fully depend on cholinergic input in the field distant from the electrode. This further highlights ENS wiring differences between the proximal and distal large intestine.

## Distribution mapping of responding neurons

Given that the proportion of hexamethonium blocked neurons (type I) is much larger in the distal colon (certainly at greater distances), we investigated whether there was a specific spatial distribution pattern along the length axis of the large intestine. To do this, we plotted the relative positions of all responding neurons on a spatial distribution map and color-coded them for their response signature (type I-V). When stimulated twice in control Krebs only, the different types of neurons were scattered throughout the network without any apparent pattern with respect to location or amplitude (*Figure 5A–B*).

In contrast, the spatial plots of the hexamethonium experiments revealed specific locations for the different responder types. Two distinct phenomena were uncovered. First, we found that apart from an increased proportion of type I blocked cells in the distal colon compared to proximal (as shown also in *Figure 3E and 4C*), the spatial distribution of these type I blocked neurons differs significantly between the two regions. In the distal colon, the type I blocked cells are uniformly spread along the longitudinal axis of the myenteric plexus (*Figure 5C–D*), while for the proximal colon almost 90% of the type I blocked neurons are located aboral to the stimulation site (*Figure 5C–D*).

Mapping the location of responding neurons also revealed that the majority (over 70% and 60% in distal and proximal colon respectively) of neurons whose amplitude was not reduced by hexamethonium (type III and IV), were located close to the electrode (in a 500 µm oral - aboral band) (*Figure 5D*), while a random distribution was observed in time control experiments.

## Morphology of responding neurons

Next, we took advantage of the distinct expression pattern of the genetically-encoded $Ca^{2+}$ indicator in enteric neurons to determine the size of their cell bodies (see Materials and methods). We found that the larger and smaller neurons were randomly scattered along the longitudinal axis of both the proximal and distal colon (see scatterplots in *Figure 5—figure supplement 1A*). Furthermore, although the size of responsive neurons varied substantially, the size of the responders in the proximal colon was on average smaller than in the distal (Prox: 173.2 ± 2.0 (SD: 74.9) vs Dis: 183.4 ± 3.1 (SD: 71.5) $µm^2$, p=0.008) (*Figure 5—figure supplement 1B*). However, when specific responder subpopulations were compared, apparent size differences were detected. First, the type I (blocked) neurons were significantly smaller in the distal than the proximal colon (Dis: 166.4 ± 5.9 vs Prox: 201.7 ± 2.6, p<0.001, *Figure 5—figure supplement 1B'*). Second, we found that in the distal colon the size of the type III (unchanged amplitude) neurons was significantly larger in the hexamethonium condition than in control (244.5 ± 13.3 vs 187.8 ± 5.4 $µm^2$, p<0.001, *Figure 5—figure supplement 1B''* blue bars). While in the proximal colon these type III neurons were even a little smaller in hexamethonium compared to the control situation (154.8 ± 5.2 vs 168.4 ± 3.2 $µm^2$, p=0.040, *Figure 5—figure supplement 1B''* red bars). When we mapped the location of the larger (>200 $µm^2$) type III (unaffected amplitude) neurons along the longitudinal axis of the proximal and distal colon, we found that in control conditions these neurons were spread out fairly uniformly in both gut regions (Suppl. *Figure 1C* dotted lines). However, in hexamethonium, these large neurons were clearly centered around the stimulation site in the distal colon, but not in the proximal (*Figure 5—*

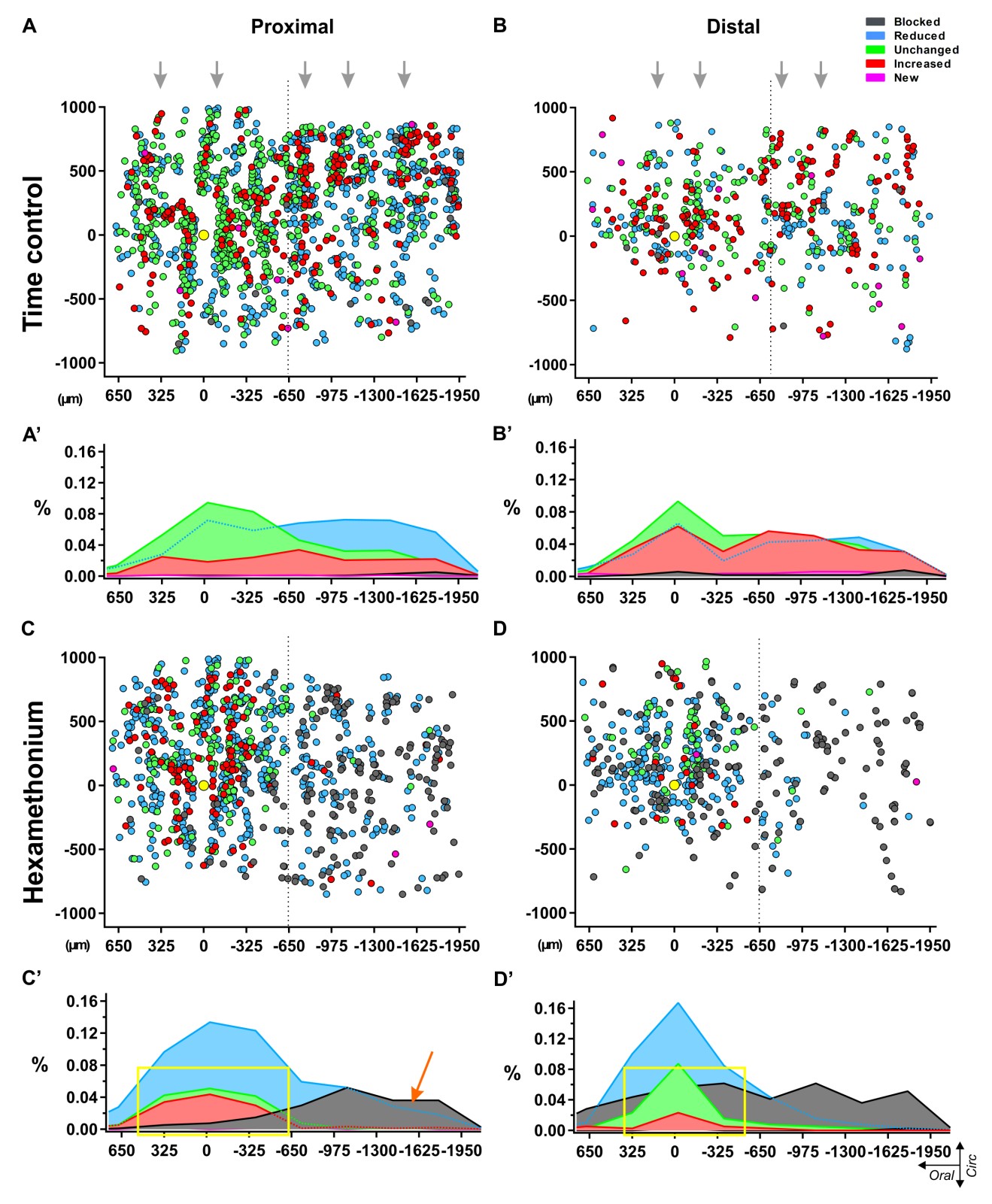

**Figure 5.** Spatial distribution of myenteric neurons responding to electrical stimulation of a single interganglionic fiber tract. Dotplots of the location of each individual neuron pooled from all recordings in control Krebs (top) (A–B) and hexamethonium (200 µM, bottom) (C–D) conditions in proximal (left) and distal (right) colon. Individual neurons are shown as circles color coded according to their response signature (=ratio of responses to two consecutive stimuli). The location of the focal stimulation electrode is indicated with the yellow circle. (A) All responsive neurons in control conditions

*Figure 5 continued on next page*

*Figure 5 continued*

were scattered without any apparent pattern, except that the ganglionic network was reflected in the distribution, indicating morphologic consistency over different preparations (grey arrows). (B) Summarizing histograms show uniform distribution of the neurons, slightly reduced (Type II: light blue) or slightly enhanced (Type IV: red), while neurons that were unchanged (Type III: green) were more centered around the electrode. Only few blocked (Type I: dark grey) and new (Type V: pink) neurons were detected in control conditions. (C) Spatial distribution of the responding cells in hexamethonium (200 µM) in proximal and distal colon. (D) Summarizing histograms show a preferential aboral location of fully blocked (Type I: dark grey) neurons in the proximal colon (orange arrow) while in the distal colon, fully blocked neurons are more spread over the entire length. Neurons showing the same (Type III) or an increased response (Type IV) cluster around the electrode in the proximal and even more so in the distal colon as indicated by the yellow box.

DOI: https://doi.org/10.7554/eLife.42914.021

The following source data and figure supplements are available for figure 5:

**Source data 1.** Spatial distribution of myenteric responding cells.
DOI: https://doi.org/10.7554/eLife.42914.026
**Figure supplement 1.** Size versus location of responding neurons.
DOI: https://doi.org/10.7554/eLife.42914.022
**Figure supplement 1—source data 1.** NO. of mice and size distribution.
DOI: https://doi.org/10.7554/eLife.42914.023
**Figure supplement 1—source data 2.** Neuronal size.
DOI: https://doi.org/10.7554/eLife.42914.024
**Figure supplement 1—source data 3.** Cumulative frequency histogram of Type III larger neurons.
DOI: https://doi.org/10.7554/eLife.42914.025

*figure supplement 1C* solid lines). This indicates that groups of neurons exist with larger cell bodies that do not depend on cholinergic transmission, these neurons operate in synchrony and are located in a defined band around the stimulus site.

## Quantification of cholinergic neurons and synaptic contacts in the proximal and distal colon

To test whether the differences in wiring are reflected in the chemical coding of both regions, we quantified the proportion of excitatory colonic myenteric neurons using immunohistochemistry against the pan-neuronal marker Hu and the excitatory cholinergic neuron marker ChAT (*Figure 6A*). Though the total number of neurons was different (see also *Figure 2*), the proportion of cholinergic neurons did not differ between the proximal and distal colon (Prox: 52.9 ± 3.9% vs Dis: 50.3 ± 4.4%, p=0.689, *Figure 6B*). Interestingly, the proportion of nitrergic neurons was found to be slightly higher in the myenteric plexus of the distal as compared to the proximal colon (Dis: 39.2 ± 1.3% vs Prox: 33.1 ± 1.3%, p=0.029, *Figure 6C*).

Since $Ca^{2+}$ imaging revealed that cholinergic transmission plays a more important role in the distal versus the proximal part of the large intestine, we investigated whether this could be reflected in the density of cholinergic synaptic contacts per neuron. We quantified the percentage of overlap between cholinergic synaptic release sites (as labeled by vAChT, *Figure 6D*) and HuC/D surfaces and found a significantly larger fraction of surface contact area between Hu+ myenteric neuronal bodies and vAChT+ cholinergic varicosities in the distal colon (Dis: 7.4 ± 0.8% vs 4.3 ± 0.4%, p=0.002, *Figure 6E*).

## Sparse labeling of neuronal projections using viral vector transduction

Since the $Ca^{2+}$ imaging data indicate that there are substantial differences in the projection patterns of neurons in the distal and proximal colon, we used viral vector transduction to sparsely label enteric neurons and track individual neuronal projections (*Figure 7A*). At two weeks post-injection, a limited number of myenteric plexus neurons along the entire colon expressed eGFP in cell bodies and fibers, which allowed us to trace their projection orientation. All traced neurons were monoaxonal and are likely motor- and interneurons (*Figure 7B*) or a subset thereof, while Dogiel type II neurons most probably escaped our labeling strategy.

We divided all eGFP-labeled neurons (57 in the proximal and 61 in the distal colon) into three groups based on their oral, aboral or circumferential projection (*Figure 7C*) as determined from the first parts of their axon. We found that significantly more neurons project aborally in the distal colon

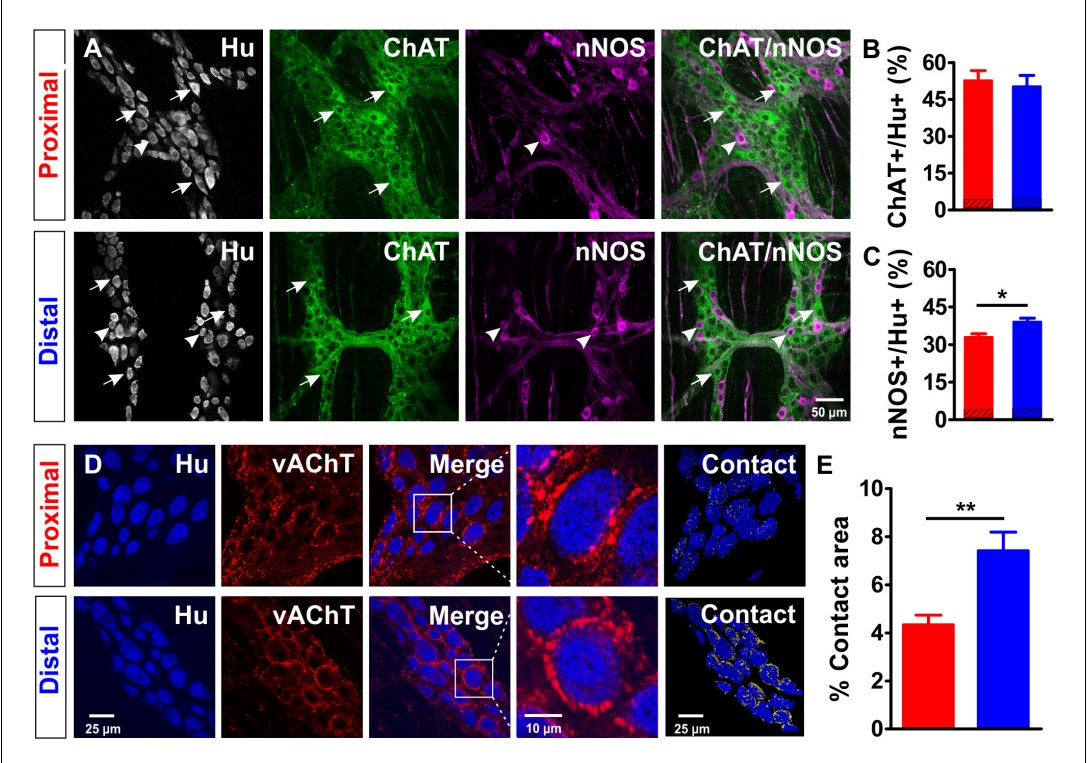

**Figure 6.** Extensive cholinergic innervation of myenteric neurons in the distal colon. (A-C) Cholinergic and nitrergic neurons in the myenteric plexus of the proximal (red) and distal (blue) colon. A, Single confocal plane of a whole-mount myenteric plexus preparation of the proximal and distal colon immunostained for Hu (magenta), choline acetyltransferase (ChAT, green) and nitric oxide synthase (nNOS, blue). Arrows and arrowheads mark some typical ChAT and nNOS neurons respectively. (B-C) Quantification of the ChAT (B) and nNOS (C) populations in the proximal (red) and distal (blue) colon, asterisks indicate statistical difference (Dis: 39.2 ± 1.3% vs Prox: 33.1 ± 1.3%, *p=0.029; N = 3). A small fraction of neurons expressed both ChAT and nNOS (3.66% and 4.00% in the proximal and distal colon, respectively), which is indicated in the dashed portions of the bars in (B) and (C). (D-E) Vesicular acetylcholine transporter (vAChT+) stainings indicate a larger contact area between cholinergic synaptic contacts and Hu+ myenteric neurons in the distal colon. (D) Single confocal plane of the myenteric plexus immunostained for Hu (blue) and vAChT (red) and its merge image. A typical neuron (white square) is shown at higher magnification. A 3D reconstruction showing the contact area (yellow) between Hu+ myenteric neuronal bodies and vAChT+ varicosities. (E) Quantification of the proportion of the surface contact area between the Hu+ myenteric neuronal bodies and vAChT + cholinergic varicosities in the proximal (red) and distal (blue) colon, asterisks indicate statistical difference (Dis: 7.4 ± 0.8% vs Prox: 4.3 ± 0.4%, **p=0.002; N = 4).

DOI: https://doi.org/10.7554/eLife.42914.027

The following source data is available for figure 6:

**Source data 1.** NO. of mice.
DOI: https://doi.org/10.7554/eLife.42914.028
**Source data 2.** Proportion of ChAT, nNOS and overlap.
DOI: https://doi.org/10.7554/eLife.42914.029
**Source data 3.** Proportion of vAChT contact area.
DOI: https://doi.org/10.7554/eLife.42914.030

compared to the proximal colon (27.3 ± 7.9% vs 11.0 ± 3.4%, p=0.004, $\chi^2$ test, *Figure 7D*). Assuming that viral transduction labels proximal and distal neurons equally, our measurements indicate that aborally projecting neurons in the distal colon are longer, while in the proximal colon the orally projecting neurons are longer. It is important to note that the projection termini could not be conclusively identified and therefore only length and not functional projection should be considered. The bimodal distribution of the neuronal projection lengths suggests that there are indeed shorter and longer projections, which can be roughly split at 3.1 mm (valley in bimodal Gaussian fit). Remarkably, this cutoff does not differ much from the subdivision based on DiI labeling made by Spencer et al. (*Spencer et al., 2005*) who suggested that the neurons with projections over 4 mm were

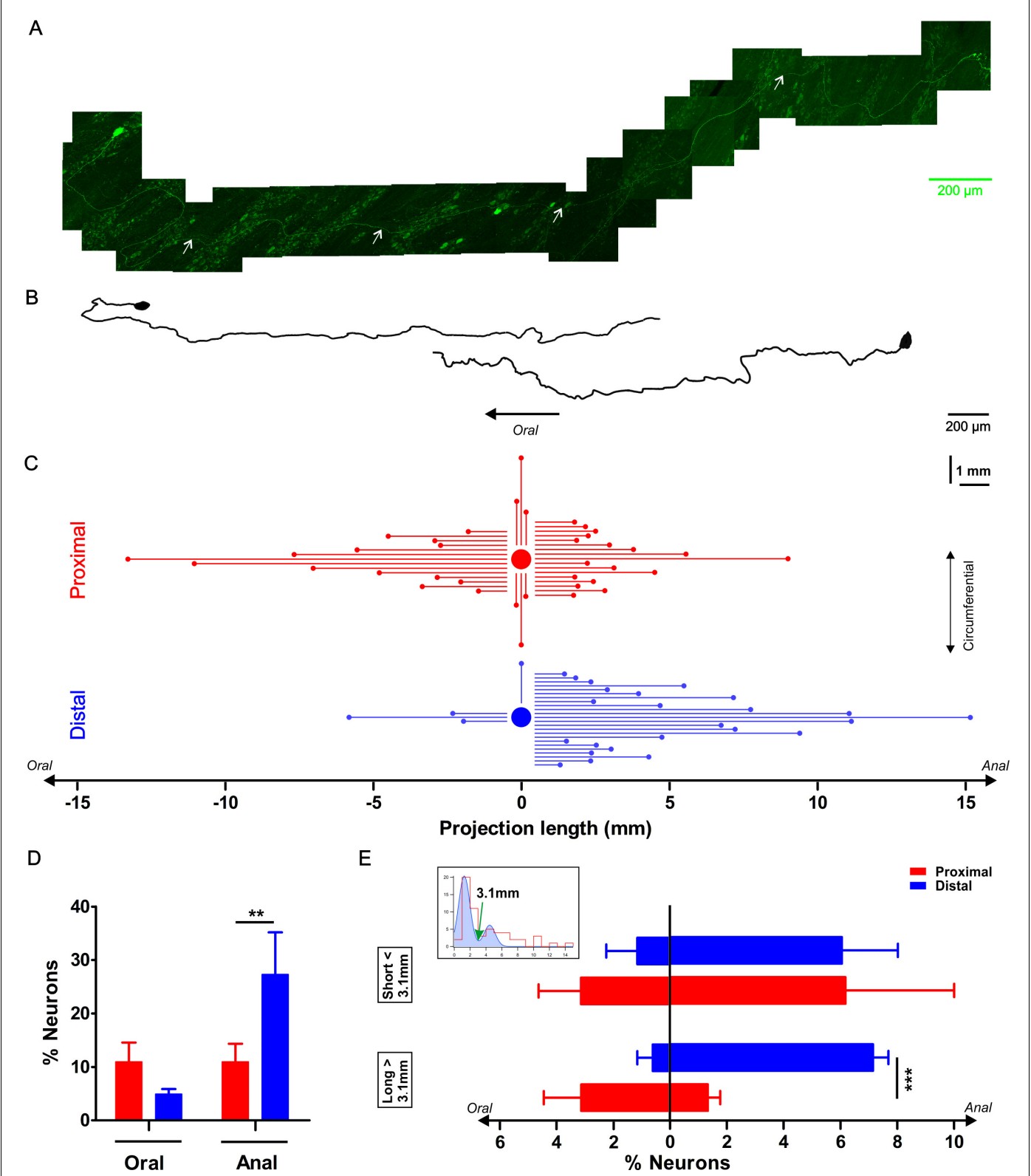

**Figure 7.** Characterization of the axonal projection length and orientation of myenteric neurons in the proximal and distal colon. (**A**) Composite image of an eGFP+ enteric neuron and its axon located in the colonic myenteric plexus of a mouse that was sparsely transduced with rAAV9-CMV-eGFP two weeks prior to tissue collection. The axon's (marked with white arrows) total length is 4.35 mm. (**B**) Two examples of reconstructed projection (one oral and one aboral) orientations of eGFP transduced myenteric neurons. (**C**) Graphic summarizing the axonal length and projection orientation of all

*Figure 7 continued on next page*

*Figure 7 continued*

tracked myenteric neurons. For about half of the eGFP expressing neurons (Prox: 37/57 and Dis: 28/61), it was possible to trace the axon to its final target. Of those, 14 projected orally (mean length of 4.6 ± 1.0 mm), 17 aborally (mean length: 2.6 ± 0.4 mm) and six circumferentially (length of 1.7 ± 0.4 mm in the proximal, while in the distal 3 projected orally (mean length: 2.9 ± 1.2 mm), 24 anally (mean length: 4.7 ± 0.7 mm) and only 1 projected circumferentially (1.4 mm). (D) Percentage of neurons projecting orally or aborally in the proximal and distal colon, asterisks indicate statistical difference (Dis: 27.3 ± 7.9% vs Prox: 11.0 ± 3.4%, p=0.004, $\chi^2$ test). (E) The inset shows the bimodal distribution of projection lengths (all pooled) and a bimodal Gaussian fit (blue) with a clear trough at 3.1 mm (green arrow). Using this value as a cutoff, the neurons were sorted in long and short projecting ones. Percentage of orally or aborally projecting neurons in the proximal and distal colon, asterisks indicate statistical difference (Dis: 7.1 ± 0.5% vs Prox: 1.3 ± 0.4%, ***p<0.001; Prox: N = 4; Dis: N = 3).

DOI: https://doi.org/10.7554/eLife.42914.031

The following source data is available for figure 7:

**Source data 1.** NO. of mice.
DOI: https://doi.org/10.7554/eLife.42914.032
**Source data 2.** Projection length.
DOI: https://doi.org/10.7554/eLife.42914.033
**Source data 3.** Percentage of neurons.
DOI: https://doi.org/10.7554/eLife.42914.034
**Source data 4.** Bimodal distribution of projection length.
DOI: https://doi.org/10.7554/eLife.42914.035

interneurons. Our data showed that there is a significant difference for longer range projecting neurons in the distal versus the proximal colon, suggesting that there is a higher proportion of descending interneurons in the myenteric plexus of the distal compared to the proximal colon (*Figure 7E*).

## Discussion

Different regions of the gut each exhibit specific motility patterns regulated by the ENS. Intestinal peristalsis is by far the best studied motor event, but little is known about how ENS circuits are differentially organized to generate regionally distinct motility patterns. In this study, we examined whether there are fundamental differences in neuronal wiring in the ENS that might reflect the capacity to initiate and control a richer portfolio of motility patterns. To do so, we designed a $Ca^{2+}$ imaging approach that allowed us to map the location and connectivity of individual neurons. Our experiments reveal that the enteric nerve circuits differ between regions of the intestinal tract and that more complex wiring is present in those regions that display more diverse motility patterns.

### Wiring complexity

The peristaltic reflex (law of the intestine) as described by *Bayliss and Starling (1899)*, experimentally confirmed by *Trendelenburg (2006)* and later refined in compartmentalized organ bath experiments (*Tonini et al., 1996*; *Spencer et al., 2001*; *Thornton and Bornstein, 2002*; *Thornton et al., 2013*; *Yuan et al., 1994*; *Johnson et al., 1998*; *Spencer et al., 2006*) has been investigated in many different studies. Despite currently available information on electrophysiological, morphological and neurochemical characteristics of enteric neurons (*Brookes, 2001*; *Furness, 2012*; *Furness, 2000*; *Schemann, 2005*) understanding of enteric circuits and actual connectivity in the ENS remains limited. This is largely because tools to probe circuits are limited, as electrophysiological recordings fail to record from many neurons simultaneously and chemical coding based on a selected number (one to three) of markers can only be applied to fixed tissues.

We used low magnification GCaMP-based $Ca^{2+}$ imaging, to include a large number of neurons in the recording field at the minor expense of losing some detail provided by higher magnification and numerical aperture lenses (*Fung et al., 2017*; *Boesmans et al., 2013*; *Foong et al., 2015*; *Hao et al., 2011*). Despite the fact that the temporal resolution of $Ca^{2+}$ recordings does not resolve individual synaptic events, the signal quality was sufficient to monitor $Ca^{2+}$ transients reliably in consecutive rounds of stimulation, which allowed us to combine functional imaging and response characterization with spatial mapping at the cellular level. Responding neurons were located in all directions surrounding the electrode, without any apparent spatial pattern. In the proximal colon, more neurons respond to electrical stimulation of interganglionic fiber tracts, which can partially be

attributed to a higher density of neurons and neuronal fibers in this gut region. However, although the mere presence of extra neurons may already reflect a more diverse set of motor patterns, it does not necessarily imply differences in neuronal wiring. To address this issue, we assumed a simple model in which one neuron has a neurite that connects to one postsynaptic neuron. An electrode that is placed on that neurite would therefore stimulate two neurons: one neuron due to neurotransmitter release (synaptically) as well as the neuron the neurite belongs to (antidromically). Simple arithmetic suggests that the distal colon matches this assumption quite well (18 (fibers) times 2 = 36, which closely approximates the observed 40 responsive neurons per field of view). However, the observed number of responding neurons in the proximal colon exceeds this prediction, suggesting that wiring is more complex in the proximal colon, with neurons connecting to multiple postsynaptic neurons (*Figure 8*).

## Circuitry probing using pharmacological inhibition and immunohistochemistry

To control for possible potentiating or inhibiting effects of our electrical stimulation protocol, we compared the $Ca^{2+}$ transients generated by two consecutive stimuli and used amplitude ratio as a fingerprint for every individual neuron's behavior in the network. Based on the limited signal to noise ratio in the low magnification recordings, we chose a rather strong stimulus to ensure that all neurons functionally connected to the stimulation site could be resolved. Because our stimulation paradigm, normally used to elicit slow excitatory postsynaptic potentials in the ENS, also releases other

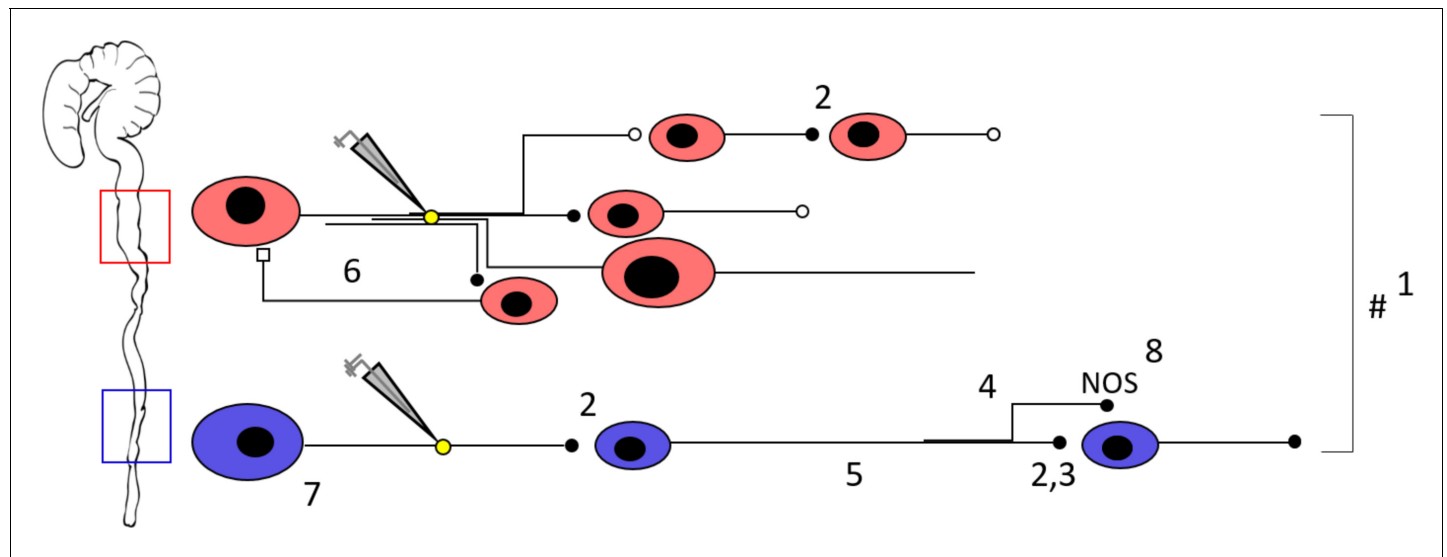

**Figure 8.** Schematic representation of the neuronal circuitry differences in the proximal versus the distal colon as determined by $Ca^{2+}$ imaging, focal electrical stimulation (represented by electrode and yellow circle), spatial analysis, viral vector tracing and immunohistochemistry. Using a 5x lens, we observed that fewer neurons (1) respond to focal electrical stimulation in the distal colon, which cannot be simply explained by the fewer number of neurons present and the fewer neuronal fibers that were stimulated. This suggests that the proximal circuitry is more complex and has more branching projections. Nicotinic transmission (represented by the black full circles, 2) plays a more important role in the distal colon (as evidenced by the effect of hexamethonium, Hex, 200 μM, and supported by VAChT immunohistochemistry) (3), a phenomenon that scales with distance from the stimulation electrode (4). AAV9 viral vector tracing indicated that there are more long distally projecting neurons in the distal colon than in the proximal colon (5). In addition, we identified an inhibitory circuit component that is dependent on cholinergic transmission (6) in the proximal colon. Neuronal size measurements reveal that Hex-independent neurons with large cell bodies are localized in the vicinity of the stimulation site in the distal colon (7). Immunohistochemical stainings showed a higher proportion of nitric oxide synthase (nNOS) neurons in the distal colon (8), these are likely to be inhibitory motor neurons and therefore endpoints of the circuitry. Taken together, the neuronal circuitry in the proximal colon is clearly more complex in its wiring compared to the distal colon. This difference in complexity may well reflect the richness of the palette of motor patterns that the specific gut regions can exert. See also *Figure 8—video 1* for an animated buildup of the schematic.
DOI: https://doi.org/10.7554/eLife.42914.036
The following video is available for figure 8:
**Figure 8—video 1.** Neuronal circuitry differences in the proximal and distal colon.
DOI: https://doi.org/10.7554/eLife.42914.037

transmitters (e.g. substance P, 5-HT, . . .), future experiments, using other specific receptor blockers, alternative electrical stimuli, or stimulation at physiological temperatures, will be necessary to refine circuitry maps.

The responses to two consecutive stimuli were robust and the distribution of response signatures (from blocked to enhanced) did not differ between the proximal and distal colon when tested in control solution. However, in hexamethonium many more neurons were completely blocked in the distal colon than in the proximal colon, indicating that transmission in the more distal region has a heavier dependence on cholinergic excitatory synaptic potentials. The majority of type I (blocked) neurons in the presence of hexamethonium was located aboral to the site of the electrode. By extending the recording one field further in the aboral direction, our data show that a greater proportion of neurons in the distal colon had responses that were abolished by hexamethonium and that this effect scales with distance from the stimulus site. This finding reiterates that neuronal wiring in the distal colon is strongly polarized and more simple than in the proximal colon, consisting of a greater number of serial monosynaptic nicotinic cholinergic neurotransmission units.

We also used immunohistochemistry to identify the cholinergic constituents (ChAT neurons and vAChT terminals) that could correlate with the functional differences observed in both regions. The proportions of cholinergic neurons were similar to those previously reported for the mouse colon (*Sang and Young, 1998*; *Hao et al., 2013*; *Erickson et al., 2014*), but in contrast to the submucous plexus (*Foong et al., 2014*), we did not detect differences in the proportion of cholinergic neurons between the proximal and distal colon. However, we found a higher percentage of nitrergic neurons in the distal versus the proximal colon. Next, we explored whether the organization of cholinergic synaptic contacts reflected the importance of cholinergic transmission in the distal compared to the proximal colon. The overwhelming abundance of vAChT positive terminals did not allow quantifying synaptic contact numbers, but the fraction of surface overlap between the vAChT and neuronal soma surfaces was significantly higher in the distal colon. This corroborates our findings derived from the functional $Ca^{2+}$ recordings that, in the distal colon, circuitry relies more importantly on cholinergic transmission. However, future experiments are necessary to investigate whether nNOS neurons (especially as they are relatively speaking more abundant) are the preferred partners of this more intense synaptic innervation. If this is the case it would explain that the important descending cholinergic component in the distal colon is essentially an inhibitory one at the level of organ function. One could also argue that this larger proportion of nitrergic neurons constitutes the nNOS + population of long descending interneurons that underlies the 'occult reflex' (*Heredia et al., 2010*; *Dickson et al., 2007*) that operates in distal colon. However, it is currently hard to predict how this mechanism (colonic elongation inhibiting pellet propulsion and CMMCs) fits our distal colon model as the blockage of nicotinic receptors by hexamethonium only rules out cholinergic input to these nitrergic interneurons, but does not affect activity of the aborally located target cells in case nitrergic interneuron processes were stimulated directly.

## Mapping of the relative location of neurons

Further analysis of the spatial distribution of responding neurons allowed us to draw four important conclusions. First, whilst there is little pure orally-projecting cholinergic transmission, in the proximal colon, about 90% of the neurons blocked by hexamethonium are found aboral to the stimulation site. This indicates that in the proximal colon, a greater proportion of neurons located aborally primarily relies on cholinergic transmission. Many other excitatory neurotransmitters are found in the ENS and likely also contribute to the communication with the postsynaptic neurons oral to the electrode.

Second, in the distal colon, neurons whose activity is abolished (type I) in the presence of hexamethonium are more randomly scattered around the electrode, indicating that cholinergic transmission is employed more generally in this region. Third, in the proximal colon, a relatively large fraction of neurons concentrated around the stimulation site showed enhanced responses in the presence of hexamethonium. This finding indicates that, at least locally, there are also nicotinic receptor dependent inhibitory pathways that act to reduce $Ca^{2+}$ responses in control conditions. Considering that $Ca^{2+}$ transients are responsible for $K^+$ ion efflux and match the slow afterhyperpolarisation (AH) in AH-type neurons (*Hillsley et al., 2000*; *Vanden Berghe et al., 2002*; *Vogalis et al., 2002*), it remains to be determined whether an increased $Ca^{2+}$ transient amplitude is a reflection of increased neuronal firing or a sign of enhanced inhibition. Last, the fact the nicotinic

blocker leaves a large fraction of neurons close to the stimulation site unaffected, indicates little or no contribution of cholinergic transmission in circumferential direction, and is very likely due to antidromic activation of intrinsic sensory neurons. For those cells whose response was only partially reduced by hexamethonium, we conclude that apart from cholinergic input they also receive input mediated by other excitatory neurotransmitters, which is consistent with previous reports (*Galligan and North, 2004*; *Johnson and Bornstein, 2004*; *Koussoulas et al., 2018*). The remaining hexamethonium-resistant responses could result from direct antidromic non-synaptic neuronal activation (*Foong et al., 2015*; *Martens et al., 2014*) or rely entirely on other transmitters. Together, this information confirms that specific spatial patterning is present in both colonic regions and suggests that the apparent 'salt and pepper' distribution of neuronal subtypes in myenteric ganglia is due to spatial overlap of functional units. In keeping with *Lasrado et al. (2017)* who showed that clonal clusters form the basis for the spatial and functional organization of the ENS in the mouse small intestine, it remains to be determined whether similar genetic lineage rules dictate ENS patterning in more distal intestinal regions.

## Projections of myenteric neurons

To address the projection orientation and length of individual neurons we used the AAV2/9 viral vector system (*Gombash et al., 2014*). Though viral vector transduction based protein expression is influenced by the choice of promoter, mouse strain and the mode and age of administration (*Gombash et al., 2014*; *Buckinx et al., 2016*; *Gombash, 2016*), our approach preferentially labels uniaxonal myenteric neurons. Labeled neurons located in the proximal colon were found to be shorter and have a greater proportion of orally-projecting fibers compared to the distal colon, suggesting the need for the myenteric circuitry to act more locally. In contrast, in the distal colon, more neurons projected aborally and for greater lengths, which corroborates the importance of expanding the recordings in the aboral direction. Although our sparse labeling strategy did not allow full characterization of neuronal projection termini, it is likely that the long distance neurons are descending interneurons (*Spencer et al., 2005*). Apart from the fact that intrinsic sensory neurons with Dogiel type II morphology escaped the viral labeling, we cannot exclude that a specific population of interneurons or motor neurons is preferentially labeled. However, given that more long distance descending neurons are present in the distal colon, it is more likely that these were targeted in the calcium imaging experiments, which agrees with the observation that hexamethonium dependent effects are more important in the distal colon as one moves further away from the stimulation site.

## Relation between neuron size, location and response fingerprint

Apart from response signature and location, the GCaMP3 recordings also inform us about neuronal soma size. One characteristic of enteric neurons that seems preserved over different species and regions is that some cells have large and smooth cell bodies (*Furness, 2012*). These neurons, which often express $Ca^{2+}$-binding proteins (calbindin or calretinin) and have Dogiel type II morphology, are associated with intrinsic sensory function, and are also termed intrinsic primary afferent neurons (*Furness, 2012*). Prior $Ca^{2+}$ imaging experiments have described that Dogiel type II neurons in the murine mouse colon receive prominent fast excitatory synaptic inputs from hexamethonium sensitive neural pathways (*Hibberd et al., 2018a*). These experiments were performed in an area of the myenteric plexus which corresponds to what we have defined in our study as part of the distal colon. Although in Hibberd et al. (*Hibberd et al., 2018a*) neuronal $Ca^{2+}$ activity was monitored in a field oral (up to 1 cm oral) to the site of stimulation, and single electrical pulses were used to evoke fast excitatory synaptic potentials, it is very likely that the vast majority of neurons with large cell bodies responding to train stimulation observed in our experiments also have Dogiel type II morphology. In line with the observation that in the mouse colon some Dogiel type II neurons receive slow synaptic transmission (*Nurgali et al., 2004*) an explicit population of the large cells in our experiments is unaffected by hexamethonium treatment.

Although these larger neurons appear randomly scattered in the network, the subgroup that is unaffected by hexamethonium in the distal colon has a defined location in a band close to the stimulation site. This suggests that in the distal colon a narrow band of circumferentially projecting neurons act together, and as suggested (*Gwynne and Bornstein, 2007*; *Thomas et al., 2004*) form a self-reinforcing network. Moreover, this confirms that also in the mouse colon, putative intrinsic

sensory neurons with Dogiel type II morphology do not communicate with each other by means of nicotinic cholinergic transmission (*Gwynne and Bornstein, 2007*). Because it is mostly those cells located remotely from the stimulation site that are inhibited in hexamethonium conditions, we also confirm that putative intrinsic primary afferent neurons indeed receive input from pathways involving nicotinic synapses (*Hibberd et al., 2018a*). Whether these neurons receive direct synaptic input via nAChR is not evident from our experiments. Assuming that the cells with larger cell bodies are indeed intrinsic sensory neurons, this finding fits the canonical model underlying peristalsis in which these cells (or a pool thereof) are responsible for reflex initiation. These findings also fit with the fact that neural peristalsis is the dominant motor pattern in the distal colon. Finally, we show that inhibition of cholinergic neurotransmission mainly abolishes the responses of small diameter neurons, which are likely to be motor neurons, and hence, do not participate in relaying nerve activity in the network as they are an exit point of the circuitry. This is also confirmed by the identification of a higher fraction of (generally small) nitrergic neurons in the distal colon, which are most probably inhibitory motor neurons responsible for descending relaxation.

## Conclusion

Although knowledge about neurochemical and neurophysiological properties of enteric neurons (*Sang and Young, 1998*; *Costa et al., 1996*; *Qu et al., 2008*) and how they communicate in neural circuits to organize intestinal motility (*Bornstein et al., 2004*; *Bornstein, 2006*) has been expanding steadily, it remains elusive how enteric neurons are organized in circuits, how they are physically built into a network and whether regional motility differences are reflected in the complexity of the underlying ENS.

In this study, we investigated whether the capacity of an intestinal region to generate a large palette of motor functions, would be reflected in the complexity of the underlying enteric nerve circuit. Using low magnification $Ca^{2+}$ response fingerprinting we show that the neuronal wiring in the regions with more diverse tasks (proximal colon) is more complex than in the distal colon, where there is one predominant motility pattern, peristalsis. The greater complexity of the proximal colon ENS, suggests a higher computational capacity, which might be necessary to regulate a set of functions specific for this region. Our study shows that motility control is hard-wired in the ENS and circuitry complexity matches the task portfolio of the specific region. Our data does not provide evidence in support nor argue against the possibility that different motility 'programs' are run in the different sections of the intestine as suggested by *Wood (2016)*, rather we show that regional differences in hardwiring exist in gut regions with different functions. In the proximal colon we have discovered an ascending inhibitory myenteric circuit that is dependent on nicotinic input. This feedback component, which is not present in the distal colon, acts fairly local (i.e. ~2 mm) and fits with the capacity of this part of the large intestine to generate mixing behavior, needed to maximize water and electrolyte absorption to begin pellet formation (*Costa et al., 2015*; *Costa et al., 2013*). Spencer et al. have described a synchronized and rhythmic (~2 Hz) neuronal firing pattern involving large populations of both excitatory and inhibitory neurons (*Spencer et al., 2018*). In their study, $Ca^{2+}$ imaging was performed on myenteric ganglia 15–30 mm oral to the terminal rectum, which corresponds to what we have defined in our manuscript as part of the distal colon. Although our findings do not argue against this observation it is currently not clear whether this firing pattern, which is associated with CMMC generation, is also apparent in the proximal colon. In line with our findings, it could well be that the greater circuit complexity in the proximal colon does not allow this kind of neuronal activity to be detected. Also the fact that serotonergic neurons, which are believed to be key for several colonic motor activities such as tonic inhibition and the initiation of CMMCs, are more numerous in the proximal colon, corresponds with our current findings indicating an increased level of circuit complexity in this part of the large intestine (*Smith and Koh, 2017*; *Okamoto et al., 2014*).

Better understanding of ENS circuits and further refinement of connectivity schemes will be necessary to fully comprehend gut function but may also help to understand GI motor disorders like pseudo-obstruction where at first glance numbers of neurons are not affected (*Avetisyan et al., 2015*; *Gershon, 2010*) but subtle wiring defects are the cause of impaired motility (*Sasselli et al., 2013*). Future studies will be required to investigate whether or not ENS wiring and the spatial location of its components can be revealed by conclusive immunohistochemical staining either for one or a combination of multiple markers. Given the overlap and repetition of circuitry units, it will not be

easy to add circuit information by overall staining techniques and it will always be necessary to complement them with spatial information obtained from local stimulation or a local (viral) tracing. Apart from immunohistochemical techniques, it may well be worthwhile to combine these with in situ hybridization approaches based on the currently emerging genetic information (*Lasrado et al., 2017*; *Zeisel et al., 2018*). Also, the use of optogenetics will be instrumental to refine our understanding of colonic enteric nerve circuits (*Boesmans et al., 2015*). This will require the exploration of novel regulatory elements to drive optogenetic tools in specific neuronal subtypes, or photomanipulation of the activity of single cells within a network (*Boesmans et al., 2017*) as opposed to bulk stimulation aiming at induction of colonic motility (*Hibberd et al., 2018b*).

## Materials and methods

### Animals

For calcium imaging, adult *Wnt1-Cre;R26R-LsL-GCaMP3* mice (short: Wnt1|GCaMP3) were used, where the genetically-encoded $Ca^{2+}$ indicator, GCaMP3, is expressed in all neural crest-derived cells, including enteric neurons and glia (*Boesmans et al., 2013*; *Zariwala et al., 2012*). Wnt1|GCaMP3 mice were bred by mating *Wnt1-Cre* mice (*Danielian et al., 1998*) with *R26R-LsL-GCaMP3* mice (also known as Ai38, purchased from The Jackson Laboratory, Bar Harbor, ME, USA, stock # 014538) (*Zariwala et al., 2012*). For viral injections, wild type C57Bl6/J mice were used. All mice were killed by cervical dislocation. All experiments were approved by the animal ethics committee of the KU Leuven guidelines for the use and care of animals.

### Calcium imaging

The entire colon was removed from adult male Wnt1|GCaMP3 mice (approx. 3 months old) and dissected in Krebs solution (containing in mM: 120.9 NaCl, 5.9 KCl, 1.2 $MgCl_2$, 2.5 $CaCl_2$, 1.2 $NaH_2PO_4$, 14.4 $NaHCO_3$, and 11.5 glucose, bubbled with 95% $O_2$-5% $CO_2$), cut along the mesenteric border and pinned flat, mucosa side up, in a dish lined with silicone elastomer (Sylgard 184; Dow Corning). The mucosa and submucous plexus were removed from the underlying smooth muscle and myenteric plexus layers. Strips of longitudinal muscle were carefully peeled off and the resultant circular muscle - myenteric plexus (CMMP) preparations were stretched over a small inox ring and immobilized by a matched rubber O-ring and placed in an organ bath (*Foong et al., 2015*; *Vanden Berghe et al., 2002*). The murine colon can be roughly divided into three segments including the proximal, mid and distal colon (*Treuting and Dintzis, 2012*). Each segment represents approximately one-third of the total colon length (*Freeling and Rezvani, 2016*). In this study, the proximal colon was defined as the portion with V-shaped ribbon mucosa 2 cm below the caecum and the distal colon, with flat mucosa, 4 cm from the caecum. Per animal, we dissected only a limited number of preparations for the proximal and the distal colon. These isolated preparations, mounted on a stainless steel ring, were considered as an independent sample of how the ENS in that region is organized.

GCaMP3 was excited at 470 nm, and its fluorescence emission was collected at 525/50 nm using a 5x objective on an upright Zeiss microscope (Axio Examiner.Z1; Carl Zeiss), equipped with a monochromator (Poly V) and cooled CCD camera (Imago QE), both from TILL Photonics. Images, 80 ms exposure each, were captured at a frame rate of 2 Hz. The tissue was constantly superfused with Krebs solution (in mM: 120.9 NaCl, 5.9 KCl, 1.2 $MgCl_2$, 2.5 $CaCl_2$, 1.2 $NaH_2PO_4$, 14.4 $NaHCO_3$, 11.5 glucose) at room temperature via a gravity-fed electronic valve system. Nifedipine (1 μM) was routinely added to the solution to prevent spontaneous muscle contraction. Electrical stimulation (ES; 300 μsec, 30 V, 20 Hz, 2 s) was delivered using a Grass S88 stimulator with SIU5 stimulus isolation unit via a focal stimulating electrode (50 μm diameter tungsten wire) placed on an interganglionic fiber tract. Within the assumption that the morphological organization of the ENS between animals (but not between proximal and distal regions) is similar, we chose to only stimulate one point instead of repositioning the electrode within one preparation. To inhibit nicotinic receptors, the tissue was superfused with hexamethonium (200 μM). Preparations were stimulated 2 or three times, 10 min apart: first in control Krebs, a second time in the presence of hexamethonium, following a 10 min drug wash-in period, and finally again in control Krebs (washout). Time controls for electrical stimulation were performed twice, 10 min apart, in control Krebs. Changes in GCaMP3 fluorescence, which reflects the intracellular $Ca^{2+}$ concentration ($[Ca^{2+}]_i$), were collected using TILLVISION software (TILL

Photonics) and analysis was performed as described previously (*Boesmans et al., 2013*) in IGOR PRO (Wavemetrics) using custom written macros. Regions of interest (ROIs) were drawn in the activity over time images and fluorescence intensity for each cell was calculated and normalized to its baseline starting value. Although the GCaMP3 fluorescence at rest was not identical between proximal and distal neurons, the difference was small enough (~1%) to assume equal GCaMP3 expression levels. Each stimulation pair (=test, Ctrl-Ctrl or Ctrl-Hex) was used to calculate the ratios in response amplitude. Each test was considered as an independent observation (n = test in *Figures 3* and *4*). Per test we obtained a histogram of how the ratios were distributed. These different histograms generated by different tests in different preparations were averaged and shown in *Figure 3*, the error bars, therefore, reflect the variation per test. The center of the ROI served to determine the location of the responding neuron relative to the stimulation electrode. To compute their size, we used the long and short axis of the ROI to calculate the surface of an ellipsoid shape with the same dimensions.

## Live video imaging of colonic motility

Ex vivo video imaging and analysis of colonic motility was performed as described previously (*Sasselli et al., 2013*; *Swaminathan et al., 2016*). Entire colons with adhering caecum were carefully isolated and loosely pinned in an organ bath chamber, continuously superfused (flow rate: 3 ml per min) with Krebs solution bubbled with carbogen (95% $O_2$ and 5% $CO_2$) and kept between 35°C and 37°C. Intestines were allowed to equilibrate, which led to the expulsion of varying amounts of luminal content. After 30 min, movies of colonic motility were captured (4 Hz frame rate, 15 min duration) with an ORCA-Flash 4.0 camera using HCImage Live software (Hamamatsu Photonics, Germany). Images were read into IGOR PRO and spatiotemporal maps were created and analyzed using custom-written algorithms.

## Neuronal process tracing

### Recombinant adeno-associated viral vector (AAV) preparation

The rAAV2/9 vector production and purification were performed by the Leuven Viral Vector Core as previously described (*Van der Perren et al., 2011*). An adeno-associated viral vector encoding the enhanced green fluorescent protein (eGFP) reporter under the ubiquitous cytomegalovirus (CMV) promoter was packaged in an AAV9-capsid. Briefly, HEK 293 T cells were transfected using a 25 kDa linear polyethylenimine solution using the pAdvDeltaF6 adenoviral helper plasmid, pAAV2/9 serotype and AAV-TF CMV-eGFP-T2A-fLuc (AAV transfer plasmid encoding eGFP and firefly luciferase reporters driven by a CMV promoter) in a ratio of 1:1:1. Viral vector particles collected from the concentrated supernatant, were purified using an iodixanol step gradient. The final sample was aliquoted and stored at - 80°C. Titers (GC/mL) for AAV stocks were analyzed by real-time PCR.

### rAAV2/9 injection

Intravenous tail vein injections of rAAV2/9-CMV-eGFP were delivered into wild type C56Bl6/J adult mice. Mice (N = 3) were placed under an incandescent lamp for 15–20 min and physically restrained. In a set of preliminary experiments, we compared several concentrations for AAV2/9-CMV-eGFP, and found that a 10–25 µl tail vein injection could sparsely transduce neurons in the mouse myenteric plexus. For the data presented in this paper, 10 µl viral particle solution (titer: $8.47 \times 10^{11}$ GC/ml) supplemented with 5% sucrose in 0.01 M PBS for a total volume of 250 µl was injected into the vein at a slight angle using a 33 gauge needle. Mice were sacrificed 2 weeks after injection and intestinal tissues were fixed, washed and prepared for immunohistochemistry.

## Immunohistochemistry

To visualize cholinergic neurons, immunohistochemistry was performed as previously described (*Boesmans et al., 2014*). Briefly, whole-mount preparations of mouse colon were pinned in a Sylgard plate containing Krebs solution continuously oxygenated with carbogen (95% $O_2$/5% $CO_2$). The mucosa and submucosal layers were dissected away and the tissue was fixed in 4% paraformaldehyde (PFA, Merck, Overijse, Belgium) in 0.1 M phosphate buffered saline (PBS, pH = 7.3–7.4) for 40 min. After washing in PBS, the longitudinal muscle layer was carefully removed to expose the myenteric plexus for later immunostaining. To visualize cholinergic neurons, the tissues were

permeabilised in 1% triton X-100 in PBS for 4–6 hr at room temperature, and incubated in primary antibodies (*Table 1*) diluted in blocking solution (PBS with 3% bovine serum albumin with 0.1% Triton X-100) for 48 hr at 4°C. To visualize AAV2/9-CMV-eGFP transduced neurons, a rat anti-GFP (*Table 1*) antibody was used. To visualize cholinergic neuronal varicosities, tissue preparations were permeabilised in 0.5% triton X-100 in PBS containing 2% donkey serum plus 2% goat serum for 2 hr at room temperature, and incubated in primary antibodies (*Table 1*) overnight at 4°C. After primary antibody labeling, all preparations were washed in PBS (3 × 10 min) and incubated in blocking solution containing matched secondary antibodies (*Table 1*) for 2 hr at room temperature.

## Image analysis

Choline acetyltransferase/neuronal nitric oxide synthase (ChAT/nNOS) and HuC/D preparations were imaged on a Zeiss LSM 780 laser scanning confocal microscope (25 x, $H_2O$ immersion lens, NA = 0.8). Cells were counted manually using ImageJ (NIH, Bethesda, MD) and the ChAT/nNOS identity of a neuron was scored using a single plane where the nucleus and cytoplasm were clearly visualized. A minimum of three fields of view in each region were analyzed for each animal and data were obtained from a minimum of three mice from three different litters. The preparations, used for visualization of cholinergic varicosities, were imaged on a Zeiss LSM 880 laser scanning confocal microscope (40x, oil immersion lens, NA = 1.3). Image stacks were deconvolved using Huygens professional (SVI, Hilversum, The Netherlands). The background fluorescence was automatically estimated and corrected for using Huygens's default parameters. Subsequently, the deconvolved image stacks were imported in IMARIS 9.02 (Bitplane, Zurich, Switzerland) to assess the surface to surface

**Table 1.** Antibodies used for immunohistochemistry.

| Antibodies | Host | Dilution | Source/Catalog number/RRID number |
|---|---|---|---|
| ChAT | Goat | 1:500 | Fisher scientific; AB144P; AB_2079751 |
| GFP | Rat | 1:1000 | Gentaur; 04404-84; AB_10013361 |
| HuCD | Human | 1:2000 | Gift from Kryzer Thomas J |
| nNOS | Sheep | 1:5000 | Gift from Miles Emson |
| Tuj1 | Rabbit | 1:2000 | Covance; PRB-435P-100; AB_291637 |
| vAChT | Guinea pig | 1:500 | Synaptic Systems; 139105; AB_10893979 |
| Anti-goat A488 | Donkey | 1:1000 | Molecular Probes; A-32814 |
| Anti-rat A488 | Donkey | 1:1000 | Molecular Probes; A-21208; AB_141709 |
| Anti-rabbit A488 | Donkey | 1:1000 | Molecular Probes; A-21206; AB_141708 |
| Anti-human A594 | Donkey | 1:1000 | Jackson Immuno Labs; 709-585-149; AB_2340572 |
| Anti-sheep A647 | Donkey | 1:500 | Molecular Probes; A-21448 |
| Anti-human AMCA | Goat | 1:250 | Jackson Immuno Labs; 109-155-003; AB_2337696 |
| Anti-guinea pig A594 | Goat | 1:500 | Molecular Probes; A-11076; AB_141930 |

Abbreviations: ChAT, choline acetyltransferase; GFP, green fluorescent peptide; nNOS, neuronal nitric oxide synthase; Tuj1: neuronal class III β-tubulin; VAChT, vesicular acetylcholine transporter.
DOI: https://doi.org/10.7554/eLife.42914.038

contact area between the Hu+ neuronal bodies and vesicular acetylcholine transporter immunoactive (vAChT+) varicosities. First, we created a surface area based on the Hu channel that was used to make a 3D mask channel for vAChT later. Then a new surface for vAChT mask channel was established. Lastly, the surface to surface contact area algorithm available in IMARIS Xtensions was applied to calculate overlap between the vAChT and HuC/D surface areas. Neuronal class III β-tubulin (Tuj1) preparations were imaged on a Zeiss LSM 780 laser scanning confocal microscope (63x, $H_2O$ immersion lens, NA = 1.15). Image stacks were deconvolved using Huygens professional (SVI, Hilversum, The Netherlands) to improve spatial resolution and fibers were counted aided by ImageJ (NIH, Bethesda, MD). To facilitate the detection of eGFP, an antibody against GFP was used. The labeled axons were traced (for at least 200 µm) to determine their individual projection orientation. In those neurons where the axon could be fully traced, we also measured the length of individual GFP-immunoreactive fibers using Image J.

## Data analysis

All data are presented as mean ±SEM. Depending on the question, 'n' refers to the number of cells or number of tests as indicated, 'N' refers to the number of animals. Actual p values (up to three decimal digits) were listed, unless they were smaller than 0.001 or they were not specified by Graphpad (as is the case for Bonferroni *post hoc* following an ANOVA test). Student's t-tests were used to compare results unless mentioned otherwise. At least three animals were used for each experimental condition. All $Ca^{2+}$ transient analysis, spatial mapping algorithms used to correlate size, position and response characteristics were custom-written in IGOR (Wavemetrics, Oregon, US) and can be found uploaded online (Source code file 1) (please check regularly for updates via www.targid.eu >LENS). Statistical analyses were performed with Microsoft Excel or GraphPad. Differences were considered to be significant if p<0.05.

## Acknowledgements

We would like to thank Joel C Bornstein from the University of Melbourne, Australia for insightful comments on the manuscript. We thank the members of LENS for their assistance during experiments and for their constructive comments on the project and manuscript.

## Additional information

### Funding

| Funder | Grant reference number | Author |
|---|---|---|
| Chinese Scholarship Council | 201408370078 | Zhiling Li |
| National Health and Medical Research Council | APP1655567 | Marlene M Hao |
| Fonds Wetenschappelijk Onderzoek | 12G1214N | Marlene M Hao |
| KU Leuven | C32/15/031 | Veerle Baekelandt |
| Fonds Wetenschappelijk Onderzoek | SBO/S006617N | Veerle Baekelandt |
| Fonds Wetenschappelijk Onderzoek | G.0921.15 | Werend Boesmans Pieter Vanden Berghe |
| Hercules Foundation | AKUL/15/37 | Werend Boesmans Pieter Vanden Berghe |
| Hercules Foundation | AKUL/11/37 | Pieter Vanden Berghe |
| Hercules Foundation | AKUL/13/37 | Pieter Vanden Berghe |
| Belgian National Fund for Scientific Research | G.0921.15 SBO/S006617N | Pieter Vanden Berghe |
| Instituut voor Innovatie door Wetenschap en Technologie | SBO/130065 | Pieter Vanden Berghe |

The funders had no role in study design, data collection and interpretation, or the decision to submit the work for publication.

## Author contributions
Zhiling Li, Data curation, Formal analysis, Validation, Investigation, Visualization, Methodology, Writing—original draft, Writing—review and editing, Collected the data, Performed the main analysis; Marlene M Hao, Conceptualization, Supervision, Investigation, Methodology, Writing—review and editing, Determined the experimental design, Contributed to the conception of the project; Chris Van den Haute, Resources, Methodology, Provided materials (viral vectors) and helped with the preliminary viral vector tracing experiments; Veerle Baekelandt, Resources, Funding acquisition, Writing—review and editing, Provided materials (viral vectors) and helped with the preliminary viral vector tracing experiments; Werend Boesmans, Conceptualization, Resources, Supervision, Funding acquisition, Validation, Investigation, Visualization, Methodology, Writing—review and editing, Determined the experimental design, Contributed to the conception of the project; Pieter Vanden Berghe, Conceptualization, Software, Supervision, Funding acquisition, Visualization, Methodology, Project administration, Writing—review and editing, Determined the experimental design, Contributed to the conception of the project, Wrote the analysis routines in Igor pro

## Author ORCIDs
Zhiling Li (iD) http://orcid.org/0000-0002-3888-971X
Marlene M Hao (iD) http://orcid.org/0000-0002-9701-8252
Veerle Baekelandt (iD) http://orcid.org/0000-0001-8966-2921
Werend Boesmans (iD) http://orcid.org/0000-0002-2426-0451
Pieter Vanden Berghe (iD) http://orcid.org/0000-0002-0009-2094

## Ethics
Animal experimentation: All experiments were approved by the animal ethics committee of the KU Leuven guidelines for the use and care of animals (specific license numbers: P192-2013; P017-2013; P021-2015)

## Decision letter and Author response
Decision letter https://doi.org/10.7554/eLife.42914.043
Author response https://doi.org/10.7554/eLife.42914.044

## Additional files

### Supplementary files
• Source code 1. Ca Imaging Analysis.
DOI: https://doi.org/10.7554/eLife.42914.039
• Source code 2. Installation instructions and user guide.
DOI: https://doi.org/10.7554/eLife.42914.040
• Transparent reporting form
DOI: https://doi.org/10.7554/eLife.42914.041

### Data availability
All data generated or analysed during this study are included in the manuscript and supporting files. Source data files have been provided for each of the figures.

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
