## [Decision Letter]

Thank you for submitting your article "Regional complexity in enteric neuron wiring reflects diversity of motility patterns" for consideration by *eLife*. Your article has been reviewed by three peer reviewers and the evaluation has been overseen by a Reviewing Editor and Eve Marder as the Senior Editor. The following individual involved in review of your submission has agreed to reveal his identity: Robert Heuckeroth (Reviewer #3).

The reviewers have discussed the reviews with one another and the Reviewing Editor has drafted this decision to help you prepare a revised submission.

Summary:

In this manuscript, the authors set out to better understand the circuitry of the enteric nervous system underlying different regions of the colon which differ in peristalsis and motility patterns. This is an important, still poorly understood area of biology that has relevance for rare disorders like chronic intestinal pseudo-obstruction syndrome and for much more common bowel motility disorders like functional constipation and irritable bowel syndrome. The authors define different motility patterns in the proximal vs. distal colon, and then use calcium imaging to show that myenteric plexus neurons in the proximal colon have a larger number of responding cells. They show that there is a higher density of neurons in the myenteric plexus in the proximal colon as compared to the distal colon, and that these neurons display slightly different response properties, specifically that neuronal connections within the distal colon are more dependent on cholinergic transmission. Finally, they use viral transduction to sparsely label neurons within enteric nervous system and find that there is a higher number of longer range projecting neurons in the distal colon than in the proximal colon. Overall, this is an interesting paper showing that there are regional differences in the connections of neurons in the myenteric plexus in different parts of the colon. Whether the distinct types of neural activity or projection patterns in proximal versus distal colon underlie differences in motility is not determined. Nevertheless, the reviewers were in agreement that this study advances our understanding of how segmental enteric neural circuitry confers regional-specific intestinal function, and therefore submission of a revised paper addressing the issues raised would be welcome.

Essential revisions:

Three major reviewer comments and required revisions are outlined here:

1) A mechanistic connection between anatomical and imaging/physiological measurements and motility differences is not made. Although the three reviewers have decided against requiring new experiments, this issue should be fully discussed in the revision. Additionally, prior findings regarding colonic motility patterns with stretch/luminal contents should be included in this discussion.

2) Data analysis and sample size issues are problematic and need to be addressed in the revision. This may require new experiments to increase sample sizes, and additional explanation/data analysis/statistics will be required.

This data analysis and sample size issue is summarized in the following points:

a) It appears that data were collected in many or all cases from only a single 5x section in the proximal and distal colon per animal.

b) "[…] we were able to record from a large population of neurons in ~29.0 ± 2.0 ganglia in a 1.3 mmx1.7 mm view". The number of samples tested here should be stated, and whether this was from proximal or distal colon.

c) (Figure 1 legend) "the average number of neurons responding per field of view (123 ± 35.3 vs 41.0 ± 4.4). […] A total of 988 neurons in the proximal (8 preparations, N=5) and 287 neurons in the distal (7 preparations, N=5) colon displayed Ca^2+^ transient". At first reading, 988 neurons sound like a lot, but 988 neurons from 8 samples equals 123.5 neurons per sample, which is the number given in the first phrase, which means that the analysis was limited to just one single field of view. Similarly in the distal colon (287/7=41, which is equal to one view).

d) (Figure 3C legend) "A total of 717 neurons in the proximal (N=4) and 252 neurons in the distal (N=4) colon displayed Ca^2+^ transient". Again, working the math here, 17/4=179.25, which is similar to the value of 123 from above, meaning that this analysis was conducted with one or perhaps 2 views per animal). For the distal segment, 252/4=63, which is close to 41, meaning that one or perhaps 2 views were used.

e) (Figure 3E legend) "A total of 653 neurons in the proximal (6 preparations, N=4) and 288 neurons in the distal (7 preparations, N=5) colon displayed Ca^2+^ transient" For the proximal, 653/6=108.83, which is equal to one view, and 288/7=41.14 for the distal, which is also equal to one view.

f) (Figure 4 legend) "A total of 284 neurons in the proximal (4 preparations, N=2) and 101 neurons in the distal (4 preparations, N=3) colon displayed Ca^2+^ transient" Here, these numbers work out to be even less than one view (284/4=71, which is about half of a view, etc.).

3) Concerns were raised about the limited advances made regarding the neuronal types and circuit components contributing to different motility patterns observed in proximal and distal colon. After discussion amongst the reviewers it was agreed that this type of detailed circuit analysis should be addressed in future studies. The issues of neuronal type identification, the use of genetic tools, and other approaches that could be used to answer outstanding questions regarding ENS circuits underlying differences in colonic motility patterns, observed in this study, should be discussed in the revision.

---

## [Author Response]

Essential revisions:Three major reviewer comments and required revisions are outlined here:1) A mechanistic connection between anatomical and imaging/physiological measurements and motility differences is not made. Although the three reviewers have decided against requiring new experiments, this issue should be fully discussed in the revision. Additionally, prior findings regarding colonic motility patterns with stretch/luminal contents should be included in this discussion.

We agree with the reviewers that our study does not provide a single neural network model that explains motility differences between the proximal and distal colon. However, and owing to our experimental approach, we identified several specific neuronal circuit differences between both gut regions and confirm a number of previous findings concerning circuit components underlying colonic motility. We have now further emphasized these novel insights in the Results and Discussion sections and included citations to several prior studies relevant to our findings.

1) Both our Ca^2+^ imaging (functional and spatial readout) and morphological (AAV9 tracing and neurochemical labeling) findings determined the distal colon to contain a basic serial nerve circuit that because of the presence of more long descending interneurons is more polarized than the proximal colon (Figure 8, nr. 4 and 5). Furthermore, since we found a larger proportion of nitrergic neurons in the distal colon, one could speculate that these constitute the nNOS+ population of long descending interneurons that underlies the ‘occult reflex’ (Heredia et al., 2010, Dickson et al., 2007) which operates in this part of the large intestine. However, it is currently hard to predict how this mechanism (colonic elongation inhibiting pellet propulsion and CMMCs) fits our distal colon model as the blockage of nicotinic receptors (by hexamethonium) only rules out cholinergic input to these nitrergic interneurons, but does not affect activity of the aborally located target cells in case nitrergic interneuron processes were stimulated directly.

2) In the proximal colon we have discovered an ascending inhibitory myenteric circuit that is dependent on nicotinic input (Figure 8, nr. 6). This feedback component acts fairly local (i.e. ~ 2mm) and is absent from the distal colon. We speculate that it serves to assist this part of the large intestine in generating mixing behavior, needed to maximize water and electrolyte absorption to begin pellet formation (Costa M et al., 2013; Costa M et al., 2015).

3) Prior Ca^2+^ imaging experiments have described that Dogiel type II neurons in the murine mouse colon receive prominent fast excitatory synaptic inputs from hexamethonium sensitive neural pathways (Hibberd et al., 2017). These experiments were performed in an area of the myenteric plexus which corresponds to what we have defined in our manuscript as part of the distal colon). Although in Hibberd et al., neuronal Ca^2+^ activity was monitored in a field oral (up to 1 cm oral to) to the site of stimulation, and single electrical pulses were used to evoke fast excitatory synaptic potentials, it is very likely that the vast majority of neurons with large cell bodies responding to train stimulation observed in our experiments also have Dogiel type II morphology. In line with the observation that in the mouse colon some Dogiel type II neurons receive slow synaptic transmission (Nurgali et al., 2004) an explicit population of the large cells in our experiments is unaffected by hexamethonium treatment, and it is mostly those cells located remotely from the stimulation site that are inhibited in hexamethonium conditions. This suggests that also in the mouse colon, putative intrinsic sensory neurons do not communicate with each other by means of nicotinic cholinergic transmission (Gwynne and Bornstein, 2007), and that they indeed receive input from pathways involving nicotinic synapses (Hibberd et al., 2017). Whether these neurons receive direct synaptic input via nAChR is not evident from our experiments.

4) In a recently published Ca^2+^ imaging study, Spencer et al., 2018, describe a synchronized and rhythmic (~2Hz) neuronal firing pattern involving large populations of both excitatory and inhibitory neurons. In this study the imaging was performed on myenteric ganglia 15-30mm oral to the terminal rectum, which corresponds to what we have defined in our manuscript as part of the distal colon. Although our findings do not argue against this observation, we have added to the Discussion section that it is currently not clear whether this firing pattern, which is associated with CMMC generation, is also apparent in the proximal colon. In line with our findings, it could well be that the greater circuit complexity in the proximal colon does not allow this kind of neuronal activity to be detected.

5) Serotonergic neurons are believed to be key for several colonic motor activities such as tonic inhibition and the initiation of CMMCs (Okamoto et al., 2014; Smith and Koh, 2017). The fact that they are more numerous in the proximal colon corresponds with our current findings indicating an increased level of circuit complexity in this part of the large intestine.

2) Data analysis and sample size issues are problematic and need to be addressed in the revision. This may require new experiments to increase sample sizes, and additional explanation/data analysis/statistics will be required.

We apologize for the fact that the description of the data and the sample sizes were not completely clear. We have amended this and added extra information to clarify how the datasets were compiled both in the Materials and methods and Results. We assume that the confusion comes from the fact that we wanted to report the total number of animals used, as well as the total number of cells included in the analysis, while the data in Figure 3 and Figure 4 were presented as distributions of neuron responses (blocked, reduced, equal, enhanced Ca^2+^ transient amplitudes) and are indeed averaged over different experimental tests (one histogram per test). Mentioning the total number of responding neurons most probably was not very informative, therefore we removed it from the main text.

Another reason for the confusion might be that depending on the question, we selected those tests that contained suitable information to answer a specific question (see Author response image 1). For instance, for Figure 1 E,F a single stimulus actually suffices to count the number of neurons that respond. Therefore the Ctrl situation of the experiments in which paired Ctrl-Ctrl as well as Ctrl-Hex stimuli were applied, could be used to compare the total numbers of responders in proximal vs. distal, while for comparing whether the [Ca^2+^]_i_ transient amplitude ratios (blocked, reduced, equal, enhanced) were dependent on drug and/or region only the tests with the correct stimulus pair (e.g. Ctrl – Hex) in the correct region could be used. Therefore some of the controls for Figure 1 and Figure 3 as well as the neuron size calculations actually come from the same recordings. We have clarified in Author response image 1 what data were generated per test and for which figure they were used. We hope this sorts out the confusion concerning our experimental approach.

**Author response image 1. respfig1:** Schematic overview of the data obtained for each single site stimulation in distal and proximal colon.

Per animal, we dissected only a limited number of preparations for the proximal and the distal colon. These isolated preparations were mounted on a stainless steel ring (as described) and were considered as an independent sample of how the ENS in that region is organized. In these preparations, we only positioned the electrode once and assumed that the anatomical organization in different mice was similar enough to average the observations over different preparations. Our data indicate that this is indeed a reasonable assumption. We stimulated only one site and did this in control (sometimes twice) and in a "drug" condition. This "drug” condition could either be Hex or control again. One test or sample is defined as a combination of Ctrl-Ctrl or Ctrl-Hex stimuli. These tests (= stimulation pair) were used to calculate the ratios in response amplitude. Each test was considered as an independent observation, while within a test the neurons technically could be compared with a paired statistical test, which we did not do as this is irrelevant for this study (it is known that Hex will block a lot of responses). Rather, we were interested in finding out how the proportion of cells with blocked, reduced, equal, enhanced [Ca^2+^]_i_ transient amplitudes would differ in the proximal and distal colon (and similarly in proximal and distal at a greater distance). Per test we obtained a number of neurons and a histogram of how the ratios were distributed. These different histograms were averaged and shown in Figure 3, the error bars, therefore, reflect the variation per test. We are convinced, based on the consistency of responders and response types that this sampling method is accurate and adequate, while at the same time considers limiting the number of experimental animals.

We include Author response image 1 as supplement to this rebuttal, which summarizes the experimental approach as well as the types of data that were obtained from each of the tests and the links to the figures in the paper. Furthermore, we have included four movies, two for the distal and two for the proximal colon, as supplementary material to Figure 1. These movies show the effect of the first and the second stimulus in control and illustrate the consistency of the response behavior, which is reflected in the Gaussian nature of the response ratio histograms in Figure 3C.

This data analysis and sample size issue is summarized in the following points:a) It appears that data were collected in many or all cases from only a single 5x section in the proximal and distal colon per animal.

As mentioned above, we dissected mostly one or two preparations in the proximal and one or two in the distal and averaged the results obtained over all preparations from different animals. Within the assumption that the morphological organization of the ENS between animals (but not between proximal and distal regions) is similar, we preferred to only stimulate one point instead of repositioning the electrode within one preparation. Although the latter would have generated more samples per preparation, it would have complicated the analysis in that some neurons, depending on the circuitry, would have received stimuli from different locations. In case of Hexamethonium, sampling more fields of view within one preparation would have been even more complicated, since we would have needed to wash Hex in and out multiple times, as matched CTRL – Hex recordings from the same location were essential to our analysis.

Based on the consistency of the Gaussian distribution of the response subtypes (eg. Figure 3C) we deem the number of tests sufficient to be used for the statistical analysis. Adding more would not alter the distributions and would therefore not be defendable from an ethical (animal use) point of view.

The averages and error bars in Figure 3 are calculated from the different histograms as outlined above. The actual numbers of tests and animals were added to the text and legend.

b) "[…] we were able to record from a large population of neurons in ~29.0 ± 2.0 ganglia in a 1.3 mmx1.7 mm view". The number of samples tested here should be stated, and whether this was from proximal or distal colon.

We amended this sentence to differentiate between proximal and distal colon. This sentence now reads: “With this imaging configuration (Figure 1B), we were able to record from a large population of neurons per field of view, containing 25 ± 2 ganglia for the proximal (from 8 myenteric plexus preparations, N = 5 animals) and 34 ± 2 ganglia (from 7 myenteric plexus preparations, N = 5 animals) for the distal colon.”

c) (Figure 1 legend) "the average number of neurons responding per field of view (123 ± 35.3 vs 41.0 ± 4.4). […] A total of 988 neurons in the proximal (8 preparations, N=5) and 287 neurons in the distal (7 preparations, N=5) colon displayed Ca^2+^ transient". At first reading, 988 neurons sound like a lot, but 988 neurons from 8 samples equals 123.5 neurons per sample, which is the number given in the first phrase, which means that the analysis was limited to just one single field of view. Similarly in the distal colon (287/7=41, which is equal to one view).

We agree that it was somewhat misleading to mention the total number of neurons. We have removed that from the paper and clearly indicate that, at least for the experiments in which we positioned the electrode in the middle of the field of view, about 125 (123 ± 35) neurons respond to the stimulation of one interconnecting strand in the proximal colon and 41 ± 4 in the distal. Immunohistochemical staining data suggest that ~1500 neurons are present in the 5x (1.3 mm * 1.7 mm = 2.2 mm2) field of view in the proximal colon and ~1300 in the distal colon, indicating that is only about 8 and 3% of the cells respond to the single point network stimulation, respectively.

It is important to note that the number of responders always signifies a response to either the first, the second or to both stimuli. Therefore the numbers of ‘responders’ included in the Hex experiments appear high, but that is because they responded in the Ctrl situation (to be blocked, or not, during the second stimulus in Hex). We hope that this helps to clarify the confusion with respect to numbers and fields of view.

d) (Figure 3C legend) "A total of 717 neurons in the proximal (N=4) and 252 neurons in the distal (N=4) colon displayed Ca^2+^ transient". Again, working the math here, 17/4=179.25, which is similar to the value of 123 from above, meaning that this analysis was conducted with one or perhaps 2 views per animal). For the distal segment, 252/4=63, which is close to 41, meaning that one or perhaps 2 views were used.

We removed the total number of neurons and use average numbers, as the ones calculated by the reviewers. In this figure legend, we also added the number of histograms (one per test), since these are the relevant numbers used for the statistical comparisons, these should have been the ones listed in the legend (n=6 tests, from N= 4 animals). The fact that the average number of responders aligns perfectly with the earlier calculations (prox: per FOV; dist: per FOV) is of course logic, as for Figure 1 the same dataset was used.

e) (Figure 3E legend) "A total of 653 neurons in the proximal (6 preparations, N=4) and 288 neurons in the distal (7 preparations, N=5) colon displayed Ca^2+^ transient" For the proximal, 653/6=108.83, which is equal to one view, and 288/7=41.14 for the distal, which is also equal to one view.

This is correct. What may be confusing here again is the fact that we have equal numbers of responding neurons in the Hex (Figure 3E) condition. However, this is not the number of neurons responding in the Hex condition only, but in the Ctrl-Hex experiment, as explained above. Because we always have a matched Ctrl stimulus per experiment we have similar numbers each time in the proximal (~120 neurons) and distal (~40 neurons). The fact that they are or are not blocked by Hex during the second stimulus does not change that number.

f) (Figure 4 legend) "A total of 284 neurons in the proximal (4 preparations, N=2) and 101 neurons in the distal (4 preparations, N=3) colon displayed Ca^2+^ transient" Here, these numbers work out to be even less than one view (284/4=71, which is about half of a view, etc.).

Indeed, and this is possible as we are further away from the stimulation site. Especially in the distal colon, less neurons respond to a stimulus. Even though there are less neurons, it is still possible to calculate the histogram of their response ratios. Here again, it is that histogram that is used as an n number to be averaged (over different tests in different preparations of different animals) and compared to the situation in the two regions, to the situation closer to the electrode and the situation in the presence of Hex.

3) Concerns were raised about the limited advances made regarding the neuronal types and circuit components contributing to different motility patterns observed in proximal and distal colon. After discussion amongst the reviewers it was agreed that this type of detailed circuit analysis should be addressed in future studies. The issues of neuronal type identification, the use of genetic tools, and other approaches that could be used to answer outstanding questions regarding ENS circuits underlying differences in colonic motility patterns, observed in this study, should be discussed in the revision.

Thank you very much for raising this point. Although we are convinced we added important extra information about the spatial organization of circuit components in the ENS (as summarized in Figure 8), we agree that this study could now be expanded by classical immunohistochemical identification of the responding neurons. However, this would be an immense task to do so in the proximal and distal colon as well as close and far away from the electrode. We thank the reviewers for their understanding that this extends well beyond the scope of the current study. Apart from the workload, we are also concerned that established and straightforward immunohistochemistry may not be sufficient to identify the behavior of a neuron as described in this study. Although immunohistochemical (IH) information has been invaluable to arrive at the current status of our understanding of the ENS, IH is also limited in that so far, and definitely, in mouse colon, no markers have been identified that match the functionality of one single population. Most probably IH approaches have to be immediately complemented by other genetic ISH (in situ hybridization)-type techniques based on currently emerging genetic information. (Zeisel et al., 2018; Lasrado et al., 2017). The use of optogenetic technology will also be instrumental to refine our understanding of ENS circuits in the colon (Boesmans et al., 2015). However, this will require the exploration of novel regulatory elements to drive optical actuators and sensors in specific neuronal subtypes, or photomanipulation of the activity of single cells (Boesmans et al., 2018) as opposed to bulk stimulation aiming at induction of colonic motility (Hibberd et al., 2018).

We have added the following paragraph in the manuscript:

“Future studies will be required to investigate whether or not ENS wiring and the spatial location of its components can be revealed by conclusive immunohistochemical staining either for one or a combination of multiple markers. […] This will require the exploration of novel regulatory elements to drive optogenetic tools in specific neuronal subtypes, or photomanipulation of the activity of single cells within a network (Boesmans et al., 2018) as opposed to bulk stimulation aiming at induction of colonic motility (Hibberd et al., 2018)”